# ENTROPY-DRIVEN DATA KNOWLEDGE DISTILLATION IN DIGRAPH REPRESENTATION LEARNING

## ABSTRACT

The directed graph (digraph), as a generalization of undirected graphs, exhibits superior representation capability in modeling complex topology systems and has garnered considerable attention in recent years. Despite the notable efforts made by existing DiGraph Neural Networks (DiGNNs) to leverage directed edges, they still fail to comprehensively delve into the abundant data knowledge concealed in the digraphs. This limitation results in sub-optimal performance and underscores the necessity of further exploring the potential correlations between the directed topology and node profiles from a data-centric perspective, thereby empowering model-centric neural networks with stronger encoding capabilities. In this paper, we propose **E**ntropy-driven **D**igraph knowl**E**dge distillatio**N** (EDEN), which can serve as a new data-centric digraph learning paradigm or a model-agnostic hot-and-plug data online knowledge distillation module for most existing DiGNNs to fully leverage informative digraphs. Specifically, EDEN first utilizes directed structural measurements from a topological perspective to construct a knowledge tree, guided by the hierarchical encoding theory. Subsequently, EDEN quantifies the mutual information of nodes from a feature perspective to further refine the knowledge flow, facilitating tree layer-wise knowledge distillation. As a general framework, EDEN also can naturally extend to undirected scenarios and demonstrate satisfactory performance. In our experiments, EDEN has been widely evaluated on 14 (di)graph datasets and across 4 downstream tasks. The results demonstrate that EDEN attains SOTA performance and exhibits strong improvement for prevalent (Di)GNNs.

## 1 INTRODUCTION

In recent years, Graph Neural Networks (GNNs) have achieved SOTA performance across various tasks including node-level Wu et al. (2019); Hu et al. (2021); Li et al. (2024b), link-level Zhang & Chen (2018); Tan et al. (2023), and graph-level Zhang et al. (2019); Yang et al. (2022). The effectiveness of GNNs stems from their capability to conduct message propagation over graphs, thereby capturing structural insights and node features. Unfortunately, most existing graph learning methods are tailored for undirected scenarios, resulting in a cascade of negative impacts in terms of:

(1) *Data-level sub-optimal representation*: Due to the complex structural patterns present in the real world, the absence of directed topology limits the captured relational information, thereby resulting in sub-optimal data representations with inevitable information loss Koke & Cremers (2023); Geisler et al. (2023); Maekawa et al. (2023); (2) *Model-level inefficient learning*: The optimization dilemma arises when powerful GNNs are applied to sub-optimal data. For instance, undirected GNNs struggle to analyze the connective rules among nodes in the entanglement of homophily and heterophily (i.e., whether connected nodes have similar features or same labels) Luan et al. (2022); Zheng et al. (2022); Platonov et al. (2023) due to neglect the valuable directed topology Rossi et al. (2023); Maekawa et al. (2023); Sun et al. (2024). This oversight compels the undirected methods to heavily rely on well-designed models or tricky theoretical assumptions to remedy the neglect of directed topology.

To break these limitations, Directed GNNs (DiGNNs) are proposed to adequately encode directed topology and node features Tong et al. (2020a); Zhang et al. (2021c); Rossi et al. (2023); Sun et al. (2024); Li et al. (2024a). Despite advancements in DiGNN design considering asymmetric topology, these methods still fail to fully explore the potential correlations between complex directed topology and node profiles. Notably, this potential correlation extends beyond directed edges and

high-order neighbors to unseen structural patterns associated with topology and node semantics. Therefore, we emphasize revealing abundant digraph knowledge from a data-centric perspective to ultimately enhance model learning. Specifically, (1) *Topology*: unlike undirected graphs, directed topology in digraphs offers node pairs or groups a more diverse range of connection patterns, implying rich structural knowledge; (2) *Feature*: digraph nodes present greater potential for more sophisticated feature knowledge when compared to nodes in the undirected graph that often present with predominant homophily Ma et al. (2021); Luan et al. (2022); Zheng et al. (2022). The two above data knowledge perspectives form the basis of digraphs and motivate our research. The core of our motivation is disentangling the complex directed structural patterns and node profiles.

To fully utilize this data-derived knowledge for the learning process, we adhere to the fundamental concept of data-centric ML and propose **E**ntropy-driven **D**igraph knowl**E**dge distillatio**N** (EDEN). Serving as a general data online knowledge distillation (KD) framework, EDEN seamlessly integrates digraph knowledge into the models to obtain the optimal embeddings for downstream tasks. Specifically, EDEN first employs directed structural measurement as a quantification metric to capture the natural evolution of topology in the digraph (*guided by the structural entropy theory*), thereby constructing a hierarchical knowledge tree (HKT) (*topology perspective*). Subsequently, EDEN further refines the HKT with fine-grained adjustments based on the mutual information (MI) of node profiles (*guided by the learning process*), regulating the knowledge flow (*feature perspective*). Building upon this, EDEN can be viewed as a new data-centric DiGNN or a hot-and-plug data online KD strategy for existing DiGNNs. For the motivation and key insights behind our use of HKT for digraph data online KD, please refer to Sec. 2.2. Notably, while we highlight the importance of EDEN in extracting intricate data knowledge in directed scenarios, it can naturally extend to undirected settings and still exhibit satisfactory performance. More details can be found in Sec. 4.1.

**Our contributions**. (1) *New Perspective*. To the best of our knowledge, EDEN is the first attempt to achieve data online KD for empowering digraph representation learning. It provides a new perspective for the digraph learning community and emphasizes the feasibility and importance of data-centric digraph mining. (2) *Unified Framework*. EDEN facilitates data-centric digraph learning through the establishment of a fine-grained HKT from both topology and feature perspectives. It contributes to discovering unseen but valuable structural patterns concealed in the digraph for improving learning efficiency. (3) *Flexible Method*. Through the personalized design, EDEN can be regarded as a new data-centric digraph learning paradigm. Furthermore, it can also serve as a model-agnostic hot-and-plug data online KD module, seamlessly integrating with existing DiGNNs to improve predictions. (4) *SOTA Performance*. Extensive experiments across a wide variety of tasks and di(graph) datasets demonstrate that EDEN consistently outperforms the best baselines (up to 3.12% higher). Moreover, it provides a substantial positive impact on prevalent (Di)GNNs (up to 4.96% improvement).

## 2 PRELIMINARIES

### 2.1 NOTATIONS AND PROBLEM FORMULATION

We consider a digraph $\mathcal{G} = (\mathcal{V}, \mathcal{E})$ with $|\mathcal{V}| = n$ nodes, $|\mathcal{E}| = m$ edges. Each node has a feature vector of size $f$ and a one-hot label of size $c$, the feature and label matrix are represented as $\mathbf{X} \in \mathbb{R}^{n \times f}$ and $\mathbf{Y} \in \mathbb{R}^{n \times c}$. $\mathcal{G}$ can be described by an asymmetrical adjacency matrix $\mathbf{A}(u, v)$. $\mathbf{D} = \mathrm{diag}\,(d_1, \cdots, d_n)$ is the corresponding degree matrix. Typical downstream tasks are as follows.

**Node-level Classification.** Suppose $\mathcal{V}_l$ is the labeled set, the semi-supervised node classification paradigm aims to predict the labels for nodes in the unlabeled set $\mathcal{V}_u$ with the supervision of $\mathcal{V}_l$.

**Link-level Prediction.** (1) Existence: predict if $(u, v) \in \mathcal{E}$ exists in the edge sets; (2) Direction: predict the edge direction of pairs of nodes $u, v$ for which either $(u, v) \in \mathcal{E}$ or $(v, u) \in \mathcal{E}$; (3) Three-class link classification: classify an edge $(u, v) \in \mathcal{E}, (v, u) \in \mathcal{E}$, or $(u, v), (v, u) \notin \mathcal{E}$.

### 2.2 HIERARCHICAL ENCODING THEORY IN STRUCTURED DATA

Inspired by the information theory of structured data Li & Pan (2016), let $\mathcal{G}$ be a real-world digraph influenced by natural noise. We define its information entropy $\mathcal{H}$ from both topology and feature perspectives, and $\mathcal{H}$ determines the true structure $\mathcal{T}$ of $\mathcal{G}$. The structured data knowledge $\mathcal{K}$ is concealed in $\mathcal{T}$. The basic assumptions based on the above definitions are as follows:

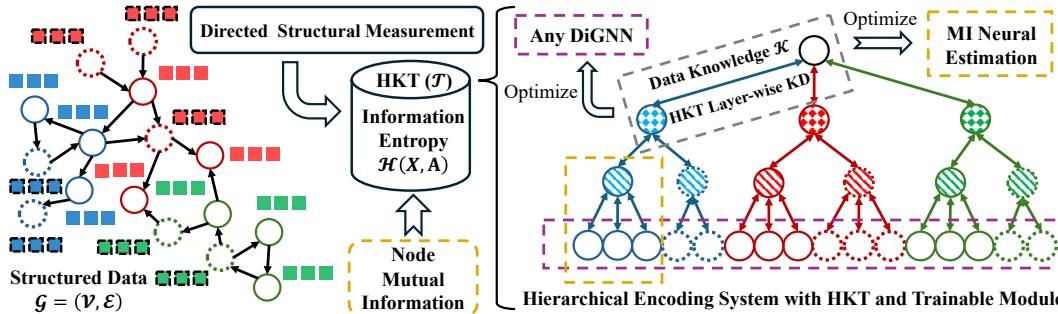

Figure 1: The overview of our proposed hierarchical encoding theory in structured data. Its core involves digraph encoding within every HKT layer and MI neural estimation across layers (illustrated using leaf nodes and their parents). Different colors and dotted lines represent distinct labels.

**Assumption 2.1.** The information entropy $\mathcal{H}$ of $\mathcal{G}$ is captured by the directed structure (topology) and trainable hierarchical encoding system (feature), reflecting the uncertainty of complex systems.

**Assumption 2.2.** The true structure $\mathcal{T}$ of $\mathcal{G}$ is obtained by excluding the maximum uncertainty $\mathcal{H}$.

**Assumption 2.3.** The knowledge $\mathcal{K}$ forms the foundation of $\mathcal{G}$ and is concealed in the true structure $\mathcal{T}$ of $\mathcal{G}$, which is used to optimize the hierarchical encoding system and achieve iterative training.

Based on these assumptions, we adhere to the hierarchical encoding theory Byrne & Russon (1998); Dittenbach et al. (2002); Clauset et al. (2008a) to establish a novel paradigm shown in Fig. 1. This paradigm standardizes the evolution of structured data in physical systems, inspiring the notion of decoding this naturally structured knowledge for analyzing complex digraphs. In other words, this trainable encoding system progressively captures the information needed to uniquely determine nodes, such as their positions, within structured data. From this, the encoded result constitutes knowledge $\mathcal{K}$ residing within the true structure $\mathcal{T}$. Subsequently, applying KD on extracted $\mathcal{K}$ from $\mathcal{T}$ optimizes the encoding system to achieve iterative training. The above concepts form the core of our motivation.

Notably, the directed structural measurement and node MI in Fig. 1 aim to uncover the structural and feature complexity of networks. Leveraging these methods, we efficiently compress information, reduce redundancy, and reveal hierarchical structures that capture subtle data knowledge often overlooked by previous studies. In other words, we minimize uncertainty and noise in $\mathcal{G}$, revealing the underlying true structure $\mathcal{T}$, which captures the layered organization of the data's inherent evolution. This $\mathcal{T}$ allows us to effectively decode the underlying knowledge $\mathcal{K}$, corresponding to the HKT in EDEN. This theoretical hypothesis has been widely applied in graph learning in recent years, driving significant research advancements in graph pooling Wu et al. (2022), contrastive learning Wu et al. (2023); Wang et al. (2023), and graph structure learning Zou et al. (2023); Duan et al. (2024).

In this paper, we adopt a data-centric perspective, which we believe has been overlooked in previous studies. Specifically, we investigate the potential of hierarchical graph data KD within the to enhance model-centric (Di)GNNs. The core intuition behind our approach is that data quality often limits the upper bound for model performance Yang et al. (2023); Zheng et al. (2023); Liu et al. (2023). By leveraging HKT, we can uncover complex patterns within graphs, enhancing data utility. This is particularly relevant for digraphs, where intricate directed causal relationships demand deeper exploration. However, our approach can also be naturally extended to undirected graphs. For further discussion on our proposed HKT and hierarchical graph clustering, please refer to Appendix A.1.

## 2.3 DIGRAPH REPRESENTATION LEARNING

To obtain node embeddings in digraphs, both spectral Zhang et al. (2021c); Lin & Gao (2023); Koke & Cremers (2023); Li et al. (2024a) and spatial Tong et al. (2020b;a); Zhou et al. (2022); Rossi et al. (2023); Sun et al. (2024) methods are proposed. Specifically, to implement spectral convolution on digraphs with theoretical guarantees, the core is to depend on holomorphic Duong & Robinson (1996) filters or obtain a symmetric (conjugated) digraph Laplacian based on PageRank Andersen et al. (2006) or magnetic Laplacian Chung (2005). Regarding spatial methods, researchers draw inspiration from the WL test Shervashidze et al. (2011) and employ message-passing mechanisms that account for directed edges. They commonly employ independently learnable weights for in- and out-neighbors to fuse node representations He et al. (2022b); Kollias et al. (2022); Sun et al. (2024).

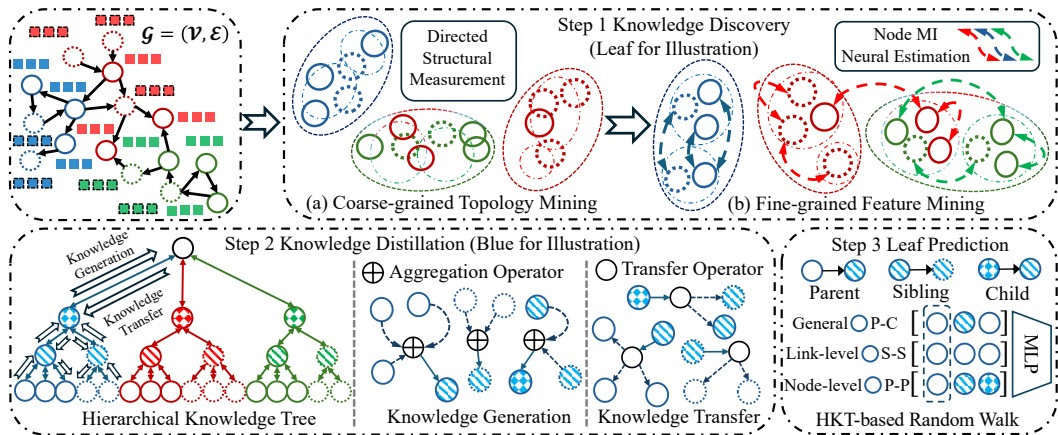

Figure 2: The overview of our proposed EDEN.

## 2.4 ENTROPY-DRIVEN MI NEURAL ESTIMATION

Information entropy originates from the practical need for measuring uncertainty in communication systems Shannon (1948). Motivated by this application, MI measures the dependence between two random variables. Based on this, Infomax Linsker (1988) maximizes the MI between inputs (features) and outputs (predictions), concentrating the encoding system more on frequently occurring patterns. To effectively estimate MI, MINE Belghazi et al. (2018) uses the DV Pinsky (1985) representation to approximate the KL divergence closely associated with MI. It achieves neural estimation of MI by parameterizing the function family as a neural network and gradually raising a tight lower bound through gradient descent. Motivated by these key insights, DGI Veličković et al. (2019) proposes graph Infomax to guide the contrastive learning process. GMI Peng et al. (2020) maximizes the MI between the current node and its neighbors, effectively aggregating features. CoGSL Liu et al. (2022) optimizes graph view generation and fusion through MI to guide graph structure learning.

## 3 METHODOLOGY

The core idea of EDEN is to fully leverage the digraph data knowledge to empower model learning. As a data online KD framework, EDEN achieves mutual evolution between teachers and students (i.e., parent and children nodes in the HKT) through the following steps as shown in Fig. 2. To avoid confusion between the data-level online KD and the widely known model-level offline KD (i.e., large teacher model and lightweight student model), we provide a detailed explanation in Appendix A.2.

*Step 1: Knowledge Discovery*: (a) To begin with, we employ directed structural measurement as a quantification metric to construct a coarse-grained HKT, discovering knowledge from a topology perspective; (b) building upon this, we perform neural estimation of node MI from a feature perspective. Through gradient descent, we regulate the coarse old knowledge flow and obtain fine-grained HKT.

*Step 2: Knowledge Distillation*: Then, we denote parent and child nodes within the same corrected partition as teachers and students to achieve data online KD. Specifically, we customize trainable knowledge generation and transfer for each parent and child by node-adaptive strategies and HKT.

*Step 3: Leaf Prediction*: Finally, we generate leaf-centered predictions (i.e., original digraph nodes) for downstream tasks. In this process, to harness rich knowledge from the HKT, we employ random walk to capture multi-level representations from their parents and siblings to improve predictions.

### 3.1 MULTI-PERSPECTIVE KNOWLEDGE DISCOVERY

In the context of digraph machine learning, original data have two pivotal components: (1) *Topology* describes the intricate connection patterns among nodes; (2) *Feature* uniquely identifies each node, closely linked to labels. If knowledge discovery focuses on only one aspect, it would lead to coarse-grained knowledge and sub-optimal distillation. To avoid this, EDEN first conducts topology mining to enrich subsequent feature mining, and collectively, establish a robust foundation for effective KD.

**Topology-aware structural measurement**. In a highly connected digraph, nodes frequently interact with their neighbors, constructing the complex topology. By employing random walks Pearson (1905), we can capture these interactions and introduce entropy as a measure of topological uncertainty Li & Pan (2016). Specifically, we can quantify one-dimensional structural information of $\mathcal{G}$ by leveraging the stationary distribution of its degrees $d$ and the Shannon entropy, which is formally defined as:

$$\mathcal{H}^1(\mathcal{G}) := -\sum_{v \in \mathcal{V}} \left( \frac{\tilde{d}_v^{\text{in}}}{m} \log \frac{\tilde{d}_v^{\text{in}}}{m} + \frac{\tilde{d}_v^{\text{out}}}{m} \log \frac{\tilde{d}_v^{\text{out}}}{m} \right), \tag{1}$$

where $\tilde{d}^{\text{in}}$ and $\tilde{d}^{\text{out}}$ are in and out-degrees of nodes in the digraph. Based on this, to achieve high-order topology mining, let $\mathcal{P} = \{\mathcal{X}_1, \mathcal{X}_2, \cdots, \mathcal{X}_{\mathcal{C}}\}$ be a partition of $\mathcal{V}$, where $\mathcal{X}$ denotes a community. To this point, we can define the two-dimensional structural information of $\mathcal{G}$ by $\mathcal{P}$ as follows:

$$\mathcal{H}^2(\mathcal{G}) = \min_{\mathcal{P}} \left\{ \mathcal{H}_{\text{in}}^{\mathcal{P}}(\mathcal{G}) + \mathcal{H}_{\text{out}}^{\mathcal{P}}(\mathcal{G}) \right\}, \ \mathcal{H}_{\text{in/out}}^{\mathcal{P}}(\mathcal{G}) :=$$

$$-\sum_{j=1}^{L} \left( \frac{\text{vol}(\mathcal{V}_j)}{m} \sum_{v \in \mathcal{V}_j} \frac{\tilde{d}_v^{\text{in/out}}}{\text{vol}(\mathcal{V}_j)} \log \frac{\tilde{d}_v^{\text{in/out}}}{\text{vol}(\mathcal{V}_j)} + \frac{g_j}{m} \log \frac{\text{vol}(\mathcal{V}_j)}{m} \right), \tag{2}$$

where $\text{vol}(\mathcal{V}) = \sum_{v \in \mathcal{V}} \tilde{d}_v^{\text{in}} / \tilde{d}_v^{\text{out}}$, $\mathcal{V}_j$ and $g_j$ are the nodes and the number of directed edges with endpoint/startpoint in the partition $j$, depend on $\mathcal{H}^{\mathcal{P}}$. Despite their effectiveness, real-world digraphs commonly exhibit a complex hierarchical structure Clauset et al. (2008b), naturally extending structural measurement to higher dimensions. Consequently, we leverage a $h$-height partition tree $\mathcal{T}$ for structured data (see Appendix A.3) to obtain $h$-dimensional structural measurement as follows:

$$\mathcal{H}^h(\mathcal{G}) = \min_{\forall \mathcal{T}: \text{Height}(\mathcal{T})=h} \left\{ \mathcal{H}_{\text{in}}^{\mathcal{T}}(\mathcal{G}) + \mathcal{H}_{\text{out}}^{\mathcal{T}}(\mathcal{G}) \right\}, \mathcal{H}_{\text{in/out}}^{\mathcal{T}}(\mathcal{G}) = -\sum_{\forall t \in \mathcal{T}, t \neq \lambda} \frac{g_t^{\text{in/out}}}{\text{vol}(\mathcal{V})} \log \frac{\text{vol}(t)}{\text{vol}(t^+)}, \tag{3}$$

where $t^+$ is the parent of $t$ and $\lambda$ is the root node of the HKT, $g_t^{\text{in}}$ and $g_t^{\text{out}}$ are the number of directed edges from other partitions to the current partition and from the current partition to other partitions, at the level where node $t$ is located. To this end, we comprehensively quantify the directed information.

**Coarse-grained HKT construction**. In contrast to the directed structural measurements defined in previous work Li & Pan (2016), EDEN addresses the limitations of forward-only random walks by incorporating reverse walks. This modification is motivated by the non-strongly connected nature of most digraphs, where the proportion of complete walk paths declines sharply after only five steps (as shown in our empirical studies in Appendix A.4). This decline suggests that most walk sequences fail to capture sufficient information beyond the immediate neighborhood of the starting node. Consequently, strictly adhering to edge directions in walks (forward-only) results in severe walk interruptions, which ultimately degrades the effectiveness of $\mathcal{H}^h(\mathcal{G})$. Furthermore, we add self-loops for sink nodes (i.e., nodes with zero in or out degrees) to prevent the scenario where the adjacency matrix might be a zero power and ensure that the sum of landing probabilities is 1. Based on this, we utilize Eq. (3) as a quantification metric and employ a greedy algorithm DeVore & Temlyakov (1996) to seek the optimal hierarchical partition tree that minimizes uncertainty. For a detailed coarse-grained HKT construction algorithm, please refer to Appendix A.5.

**Feature-oriented node measurement**. At this point, we have simulated the natural evolution of a digraph from a topology perspective, guided by the principle of minimizing directed structural uncertainty. However, as previously pointed out, node features play an equally pivotal role in digraph learning, which means that the topological measurement alone is insufficient to accurately reflect the true structure in digraphs, and mislead knowledge generation. Therefore, we aim to fully leverage node features based on the original partitions to fine-tune the coarse HKT for the subsequent KD.

The key insight of HKT refinement is to emphasize high feature similarity within the same partition while ensuring differences across distinct partitions. This is to retain authority in the parent nodes (teachers) during knowledge generation and avoid the reception of misleading knowledge by the child nodes (students). To achieve our targets, we introduce intra- and inter-partition node MI neural estimation from a feature perspective. Specifically, the former retains nodes with higher MI within the current partition. These nodes not only serve as effective representations of the current partition but also inherit partition criteria based on structural measurement. The latter identifies nodes in other partitions that effectively represent their own partitions while exhibiting high MI with the current partition. We can reserve and utilize affiliations of these nodes to improve the HKT structure.

**Partition-based MI neural estimation**. Before introducing our method, we provide a formalized definition as follows. For current partition $\mathcal{X}_p$, we first sample a subset $\Omega_p$ consisting of $K_p$ nodes from $\mathcal{X}_p$ and other partitions $\mathcal{X}_q$ at the same HKT height (more details can be found in Appendix A.6). Then, we employ a criterion function $C(\cdot)$ to quantify the information of $\Omega_p$, aiming to find the most informative subset for generating knowledge about $\mathcal{X}_p$ by solving the problem $\max_{\Omega_p \subset \mathcal{V}} C(\Omega_p)$, subject to $|\Omega_p| = K_p$. In our implementation, we formulate $C(\Omega_p)$ for $\mathcal{X}_p$ based on the neural MI estimator between nodes and their generalized neighborhoods, capturing the neighborhood representation capability of nodes. Based on this, we derive the following theorems related to MI neural estimation for structured data, guiding the design of a criterion function for HKT partitions.

**Theorem 3.1.** *Let $\mathcal{T}$ be the HKT in a digraph $\mathcal{G} = (\mathcal{V}, \mathcal{E})$. For any selected node $v \in \mathcal{X}_p$ and $u \in \mathcal{X}_q$ in the subset $\Omega_p$, we define their generalized neighborhoods as $\mathcal{N}_v^{\mathcal{T}} = \mathcal{X}_p$ and $\mathcal{N}_u^{\mathcal{T}} = \mathcal{X}_p \cup \mathcal{X}_q$. Given $v$ and $\mathcal{N}_v^{\mathcal{T}}$ as an example, consider random variables $f_v$ and $f_{\mathcal{N}_v^{\mathcal{T}}}$ as their unique node (sets) features, the lower bound of MI between $v$ and its generalized neighborhoods is given by the KL divergence between the joint distribution $P\left(f_v, f_{\mathcal{N}_v^{\mathcal{T}}}\right) = P\left(f_v = \mathbf{X}_v, f_{\mathcal{N}_v^{\mathcal{T}}} = \mathbf{X}_{\mathcal{N}_v^{\mathcal{T}}}\right)$ and the product of marginal distributions $P_{f_v} \otimes P_{f_{\mathcal{N}_v^{\mathcal{T}}}}$ can be formally defined as follows:*

$$
\begin{aligned}
\mathcal{I}^{(\Omega)}(f_v, f_{\mathcal{N}_v^{\mathcal{T}}}) &= \mathcal{D}_{\mathrm{KL}}\left(P\left(f_v, f_{\mathcal{N}_v^{\mathcal{T}}}\right) \| P_{f_v} \otimes P_{f_{\mathcal{N}_v^{\mathcal{T}}}}\right) \\
&\geq \sup_{F \in \mathcal{F}} \left\{ \mathbb{E}_{\mathbf{X}_v, \mathbf{X}_{\mathcal{N}_v^{\mathcal{T}}} \sim P\left(f_v, f_{\mathcal{N}_v^{\mathcal{T}}}\right)}\left[F\left(\mathbf{X}_v, \mathbf{X}_{\mathcal{N}_v^{\mathcal{T}}}\right)\right] \mathbb{E}_{\mathbf{X}_v \sim P_{f_v}, \mathbf{X}_{\mathcal{N}_{\bar{v}}^{\mathcal{T}}} \sim P_{f_{\mathcal{N}_v^{\mathcal{T}}}}}\left[e^{F\left(\mathbf{X}_v, \mathbf{X}_{\mathcal{N}_{\bar{v}}^{\mathcal{T}}}\right) - 1}\right] \right\},
\end{aligned}
\tag{4}
$$

*where $\bar{v}$ represents the randomly selected node in $\Omega_p$ except for $v$. This lower bound is derived from the $f$-divergence representation based on KL divergence. $\mathcal{F}$ is an arbitrary function that maps a pair of the node and its generalized neighborhoods to a real value, reflecting the dependency.*

**Theorem 3.2.** *The lower bound in Theorem 3.1 can be converted to f-divergence representations based on non-KL divergence. This GAN-like divergence for structured data is formally defined as:*

$$
\begin{aligned}
\mathcal{D}_{\mathrm{KL}}\left(P\left(f_v, f_{\mathcal{N}_v^{\mathcal{T}}}\right) \| P_{f_v} \otimes P_{f_{\mathcal{N}_v^{\mathcal{T}}}}\right) &\sim \mathcal{I}_{\mathrm{GAN}}^{(\Omega)}(f_v, f_{\mathcal{N}_v^{\mathcal{T}}}) \\
&\geq \sup_{F \in \mathcal{F}} \left\{ \mathbb{E}_{P\left(f_v, f_{\mathcal{N}_v^{\mathcal{T}}}\right)}\left[\log \sigma\left(F\left(\mathbf{X}_v, \mathbf{X}_{\mathcal{N}_v^{\mathcal{T}}}\right)\right)\right] \mathbb{E}_{P_{f_v}, P_{f_{\mathcal{N}_v^{\mathcal{T}}}}}\left[\log\left(1 - \sigma\left(F\left(\mathbf{X}_v, \mathbf{X}_{\mathcal{N}_{\bar{v}}^{\mathcal{T}}}\right)\right)\right)\right] \right\},
\end{aligned}
\tag{5}
$$

*where $\sigma(\cdot)$ is the activation function. Since solving $\mathcal{I}_{\mathrm{GAN}}^{(\Omega)}$ across the entire function space $\mathcal{F}$ is practically infeasible, we employ a neural network $F_w(\cdot, \cdot)$ parameterized by $w$.*

**Theorem 3.3.** *Through the optimization of $w$, we obtain $C(\Omega) = \widehat{I}_{\mathrm{GAN}}^{(\Omega)}$ as the GAN-based node MI neural estimation for every partition within fine-grained HKT:*

$$
\max_w \frac{1}{|\Omega|} \sum_{v \in \Omega} \log \sigma\left(F_w\left(\mathbf{X}_v, \mathbf{X}_{\mathcal{N}_v^{\mathcal{T}}}\right)\right) + \max_w \frac{1}{|\Omega|^2} \sum_{(v, \bar{v}) \in \Omega} \log\left(1 - \sigma\left(F_w\left(\mathbf{X}_v, \mathbf{X}_{\mathcal{N}_{\bar{v}}^{\mathcal{T}}}\right)\right)\right).
\tag{6}
$$

*The two terms capture the dependency and difference between selected nodes and their neighborhoods.*

**Fine-grained HKT correction**. Based on the above theorems, we instantiate the intra-partition MI:

$$
F_w^{intra} := \mathcal{Q}_{intra}\left(\mathcal{W}_1\left(\mathcal{M}\left(\mathbf{X}_v\right)\right), \mathcal{W}_2\left(\mathcal{M}\left(\mathbf{X}_{\mathcal{N}_v^{\mathcal{T}}}\right)\right)\right),
\tag{7}
$$

where $\mathcal{Q}_{intra}$ is an embedding function designed to quantify node MI by maximizing intra-partition $\mathcal{X}_p$ similarity, $\mathcal{M}$ is a model-agnostic digraph learning function, and $\mathcal{W}_1$ and $\mathcal{W}_2$ are embedding functions for selected nodes and their generalized neighborhoods. Building upon this, we extend Eq. (7) to the inter-partition scenario, enabling the discovery of potential nodes $u$ that exhibit high MI with $\mathcal{X}_p$ and inherit the directed structure measurement criteria of $\mathcal{X}_q$:

$$
F_w^{inter} := \mathcal{Q}_{inter}\left(\mathcal{W}_1\left(\mathcal{M}\left(\mathbf{X}_u\right)\right), \mathcal{W}_2\left(\mathcal{M}\left(\mathbf{X}_{\mathcal{N}_u^{\mathcal{T}}}\right)\right)\right).
\tag{8}
$$

Notably, the above equations share $\mathcal{W}_1$ and $\mathcal{W}_2$, as they are both used for encoding the current node and corresponding generalized neighborhoods. In our implementation, $\mathcal{Q}$ and $\mathcal{W}$ are instantiated as MLP and the linear layer. Furthermore, we combine it with Sec. 3.2 to reduce complexity. Detailed proofs of the theorems and discussions can be found in Appendix A.6-A.8.

## 3.2 Node-adaptive Knowledge Distillation

**Knowledge Generation**. After considering the distinctness of nodes, we obtain $\Omega_p$ for the current partition $\mathcal{X}_p$ by solving Eq. (6), where $\Omega_p$ comprises $K$ nodes selected from $\mathcal{X}_p$ and other partitions $\mathcal{X}_q$. Now, we compute an affinity score for each sampling node in $\Omega_p$ based on their unique roles $v_x$ given by coarse-grained HKT, where $v_1$ is the nodes from the current partition, and $v_2$ is the nodes obtained by performing partition-by-partition sampling of the other partitions. The sampling process is limited by the number of nodes in $\mathcal{X}_p$. The above process in the can $\mathcal{X}_p$ be formally defined as:

$$
\begin{aligned}
\mathcal{S}_{v_1} &= \sigma(\mathcal{Q}_{intra}(\mathcal{W}_1(\mathcal{M}(\mathbf{X}_{v_1})), \mathcal{W}_2(\mathcal{M}(\mathbf{X}_{\mathcal{N}_{v_1}^{\mathcal{T}}})))), \\
\mathcal{S}_{v_{2,1}} &= \sigma(\mathcal{Q}_{inter}(\mathcal{W}_1(\mathcal{M}(\mathbf{X}_{v_2})), \mathcal{W}_2(\mathcal{M}(\mathbf{X}_{\mathcal{N}_{v_2}^{\mathcal{T}}})))), \\
\mathcal{S}_{v_{2,2}} &= \sigma(\mathcal{Q}_{intra}(\mathcal{W}_1(\mathcal{M}(\mathbf{X}_{v_2})), \mathcal{W}_2(\mathcal{M}(\mathbf{X}_{\mathcal{X}_q^{v_2}})))),
\end{aligned}
\tag{9}
$$

where $\mathcal{S}_{v_1}$ and $\mathcal{S}_{v_{2,1}}$ are used to discover the knowledge closely related to the current partition. However, this strategy often causes an over-fitting problem. Therefore, we introduce $\mathcal{S}_{v_{2,2}}$ to bring diverse knowledge from other partitions. Specifically, we aim to identify and emphasize nodes that, while representing other partitions, exhibit significant differences from the current partition by $\mathcal{S}_{v_2} = \max(\mathcal{S}_{v_{2,1}}, \mathcal{S}_{v_{2,2}})$. Finally, we obtain the parent representation of $\mathcal{X}_p$ by $\mathbf{X}_p = \mathcal{S}_{\Omega_p} \mathbf{X}_{\Omega_p}$.

**Knowledge Transfer**. In this section, we introduce the personalized knowledge transfer from the parent node $\mathbf{X}_p$ (teacher) to the child nodes $\mathbf{X}_v$ (student) under partition $\mathcal{X}_p$. The key insights of our proposed node-adaptive strategy are as follows: (1) For parent nodes, not all knowledge is clearly expressible, implying that class knowledge hidden in embeddings or soft labels may be ambiguous. (2) For child nodes, each node has a unique digraph context, causing various knowledge requirements. Building upon this, our proposed strategy considers the trade-off between the knowledge held by the parent node and the specific requirements of individual child nodes, facilitating personalized transfer.

Specifically, we first refine the knowledge hidden in the parent node representation through entropy-driven $\mathcal{Q}_{parent}$ to improve knowledge quality. Then, we aim to capture the diverse requirements of child nodes in knowledge transfer by $\mathcal{Q}_{child}$ to achieve personalized transfer. Similar to Sec. 3.1, we employ MLP to instantiate $\mathcal{Q}$. To this point, we have built an end-to-end online KD framework for the mutual evolution of teacher and student by the node-adaptive KD loss, which is defined as:

$$
\mathcal{L}_{kd} = \|\mathbf{X}_p/\mathcal{U}_p - \mathcal{Q}_{child}\left(\mathbf{X}_{v_1, v_2}^{\mathcal{X}_p}\right)\|_F, \quad \mathcal{U}_p = \sigma\left(\mathcal{Q}_{parent}\left(-\sum_{i=1}^{c} \mathbf{X}_{p,i} \log\left(\mathbf{X}_{p,i}\right)\right)\right).
\tag{10}
$$

## 3.3 Random walk-based Leaf Prediction

Now, we have obtained representations for all nodes in the HKT. Then, our focus shifts to generating leaf-centered predictions for various downstream tasks. To improve performance, a natural idea is to leverage the multi-level representations, including siblings and higher-level parents of the current leaf node, to provide a more informative context. Therefore, we employ the tree-based random walk to obtain this embedding sequence. However, given a receptive field, the number of paths is greater than the number of nodes, employing all paths becomes impractical, especially with a large receptive field. To gather more information with fewer paths in the search space, we define walk rules based on the specific downstream task. Specifically, we concentrate on sampling siblings ($s_{rw}$) to capture same-level representation for link-level tasks. Conversely, for node-level tasks, we prioritize sampling from parents ($p_{rw}$) or children ($c_{rw}$) to acquire multi-level representations. Consider a random walk on edge $e_{t,s}$, currently at node $s$ and moving to the next node $r$. The transition probability is set as follows:

$$
\mathcal{P}_{rw}(v_i = r \,|\, v_{i-1} = s, v_{i-2} = t) = \begin{cases} 1/p_{rw}, \text{parent} \\ 1/s_{rw}, \text{sibling} \\ 1/c_{rw}, \text{child} \\ 0, \text{otherwise} \end{cases}.
\tag{11}
$$

Then, we concat the $k$-step random walk results (i.e., node sequence) to obtain $\mathcal{P}_{rw}^k$ for each leaf node. After that, the leaf-centered prediction and overall optimization with $\alpha$-flexible KD and MLP instantiated $\mathcal{Q}_{rw}$ are formally defined as (please refer to Appendix A.9 for complexity analysis):

$$
\mathcal{L} = \mathcal{L}_{\text{cross-entropy}}\left(\hat{\mathbf{Y}}, \mathbf{Y}\right) + \alpha\mathcal{L}_{kd},
$$
$$
\hat{\mathbf{Y}}(v) = \text{Softmax}\left(\mathcal{Q}_{rw}^{node}\left(\mathcal{P}_{rw-v}^k\right)\right), \hat{\mathbf{Y}}(u, v) = \text{Softmax}\left(\mathcal{Q}_{rw}^{link}\left(\mathcal{P}_{rw-u}^k || \mathcal{P}_{rw-v}^k\right)\right).
\tag{12}
$$

## 3.4 LIGHTWEIGHT EDEN IMPLEMENTATION

As a data-centric framework, EDEN effectively implements hierarchical digraph data online KD driven by HKT and trainable modules, while seamlessly integrating with any model-centric neural network. This framework offers new insights and tools for advancing data-centric digraph learning. However, scalability remains a bottleneck in our approach, and we aim to propose feasible solutions to enhance its efficiency. Specifically, we implement a lightweight EDEN as outlined below.

**Lightweight Coarse-grained HKT Construction**. As detailed in Algorithm 1-2 of Appendix A.5, we can introduce randomness using Monte Carlo methods, which select potential node options rather than optimal ones for sampling before executing the detaching and merging process. Probabilities are assigned to these choices, and a random option is selected for execution. This approach involves running multiple Monte Carlo simulations, where nodes are randomly chosen in each run to generate various candidate solutions. An optimal or near-optimal solution is then selected from these.

**Lightweight Fine-grained HKT Construction**. For node MI neural estimation, computational efficiency can be further optimized using techniques such as incremental training and prototype representation for label-specific children and parent nodes. This training and embedding representation method will significantly reduce the computational overhead of node MI neural estimation.

**Lightweight Layer-wise Digraph Learning Function**. We can obtain node representations through weight-free feature propagation, a computationally efficient embedding method that has proven effective in recent studies Wu et al. (2019); Zhang et al. (2022); Li et al. (2024b). Through this design, we significantly reduce the number of learnable parameters and achieve efficient gradient updates.

## 4 EXPERIMENTS

In this section, we aim to offer a comprehensive evaluation and address the following questions: **Q1**: How does EDEN perform as a new data-centric DiGNN? **Q2**: As a hot-and-plug data online KD module, what is its impact on the prevalent (Di)GNNs? **Q3**: If EDEN is effective, what contributes to its performance? **Q4**: What is the running efficiency of EDEN? **Q5**: How robust is EDEN when dealing with hyperparameters and sparse scenarios? To maximize the usage for the constraint space, we will introduce datasets, baselines, and experiment settings in Appendix A.10-A.13.

## 4.1 PERFORMANCE COMPARISON

**A New Digraph Learning Paradigm.** To answer **Q1**, we present the performance of EDEN as a new data-centric DiGNN in the Table 1 and Table 2. These tables provide a comprehensive evaluation of EDEN's performance across four downstream tasks on digraph datasets with three evaluation metrics. According to reports, EDEN consistently achieves state-of-the-art performance across all scenarios. Specifically, compared to various methods that intermittently achieve the second-best results, EDEN attains

Table 1: Test accuracy (%) in directed Node-C.

| Models | CoraML | CiteSeer | WikiCS | Tolokers | Empire | Rating | Arxiv |
|---|---|---|---|---|---|---|---|
| GCNII | 80.8±0.5 | 62.5±0.6 | 78.1±0.3 | 78.5±0.1 | 76.3±0.4 | 42.3±0.5 | 65.4±0.3 |
| GATv2 | 81.3±0.9 | 62.8±0.9 | 78.0±0.4 | 78.8±0.2 | 78.2±0.9 | 43.8±0.6 | 66.7±0.3 |
| AGT | 81.2±0.8 | 62.9±0.8 | 78.3±0.3 | 78.5±0.2 | 77.6±0.7 | 43.6±0.4 | 66.2±0.4 |
| DGCN | 82.2±0.5 | 63.5±0.7 | 78.4±0.3 | 78.7±0.3 | 78.7±0.5 | 44.7±0.6 | 66.9±0.2 |
| DIMPA | 82.4±0.6 | 64.0±0.8 | 78.8±0.4 | 78.9±0.2 | 79.0±0.6 | 44.6±0.5 | 67.1±0.3 |
| D-HYPR | 82.7±0.4 | 63.8±0.7 | 78.7±0.2 | 79.2±0.2 | 78.8±0.5 | 44.9±0.5 | 66.8±0.3 |
| DiGCN | 82.0±0.6 | 63.9±0.5 | 79.0±0.3 | 79.1±0.3 | 78.4±0.6 | 44.3±0.7 | 67.1±0.3 |
| MagNet | 82.2±0.5 | 64.2±0.6 | 78.9±0.2 | 79.0±0.2 | 78.8±0.4 | 44.7±0.6 | 67.3±0.3 |
| HoloNet | 82.5±0.5 | 64.1±0.7 | 79.2±0.3 | 79.4±0.2 | 78.7±0.5 | 44.5±0.6 | 67.5±0.2 |
| **EDEN** | **84.6±0.5** | **65.8±0.6** | **81.4±0.3** | **81.3±0.2** | **81.1±0.6** | **46.3±0.4** | **69.7±0.3** |

improvements of 2.78% and 2.24% on node- and link-level tasks. Based on Sec. 3.4, the design details of the HKT layer-wise digraph learning function in EDEN can be found in Appendix A.11.

**A Hot-and-plug Online KD Module.** Subsequently, to answer **Q2**, we present performance gains achieved by incorporating EDEN as a hot-and-plug module into existing methods in Table 3 (deployment details can be found in Appendix A.11.). Based on the results, we observe that EDEN performs better on digraphs and DiGNNs compared to undirected ones. This is because more abundant data knowledge is inherent in digraphs, coupled with the theoretically stronger representational power of DiGNNs (see Sec. 2.2). EDEN is designed to meet this specific demand, thus showcasing superior performance. Notably, the performance of EDEN as a hot-and-plug module exceeds its performance as a self-reliant method in some cases. This is attributed to the adoption of a lightweight HKT construction and layer-wise digraph learning function design for running efficiency. While this approach sacrifices some accuracy, it significantly enhances scalability for handling large-scale WikiTalk.

Table 2: Model performance (%) in three directed link-level downstream tasks.

| Datasets ($\rightarrow$) | Slashdot | | | | | WikiTalk | | | | |
|---|---|---|---|---|---|---|---|---|---|---|
| Tasks ($\rightarrow$) | Existence | | Direction | | Link-C | Existence | | Direction | | Link-C |
| Models ($\downarrow$) | AUC | AP | AUC | AP | ACC | AUC | AP | AUC | AP | ACC |
| GCN | 88.4±0.1 | 88.6±0.1 | 90.1±0.1 | 90.2±0.1 | 83.8±0.2 | 92.4±0.1 | 92.3±0.0 | 86.5±0.2 | 87.1±0.1 | 84.6±0.2 |
| GAT | 88.1±0.2 | 88.4±0.1 | 90.4±0.2 | 90.5±0.1 | 83.5±0.3 | OOM | OOM | OOM | OOM | OOM |
| OptBG | 88.6±0.1 | 88.5±0.0 | 89.8±0.1 | 90.6±0.0 | 83.7±0.2 | 92.7±0.1 | 92.2±0.1 | 87.2±0.1 | 87.3±0.1 | 85.1±0.2 |
| NAG | 88.9±0.1 | 89.1±0.1 | 90.6±0.2 | 90.4±0.1 | 84.0±0.3 | OOM | OOM | OOM | OOM | OOM |
| NSTE | 90.6±0.1 | 90.8±0.0 | 92.2±0.1 | 92.4±0.0 | 85.4±0.2 | 94.4±0.1 | 94.6±0.1 | 90.7±0.1 | 90.0±0.0 | 90.4±0.1 |
| Dir-GNN | 90.4±0.1 | 90.5±0.0 | 92.0±0.1 | 91.8±0.1 | 85.2±0.2 | 94.7±0.2 | 94.3±0.1 | 90.9±0.1 | 90.3±0.1 | 90.6±0.2 |
| MagNet | 90.3±0.1 | 90.2±0.1 | 92.2±0.2 | 92.4±0.1 | 85.3±0.1 | OOM | OOM | OOM | OOM | OOM |
| MGC | 90.1±0.1 | 90.4±0.0 | 92.1±0.1 | 92.3±0.1 | 85.0±0.1 | 94.5±0.1 | 94.2±0.0 | 90.6±0.1 | 90.2±0.0 | 90.1±0.1 |
| EDEN | **91.8±0.1** | **92.0±0.0** | **93.3±0.1** | **93.1±0.0** | **87.1±0.2** | **95.4±0.1** | **95.8±0.1** | **91.5±0.0** | **91.7±0.1** | **91.0±0.1** |

Table 3: Node-C test accuracy (%) gains brought by EDEN in Di(GNNs) under Di(graphs).

| Models | CoraML | CiteSeer | WikiCS | Arxiv | Photo | Computer | PPI | Flickr | Improv. |
|---|---|---|---|---|---|---|---|---|---|
| OptBG | 81.5±0.7 | 62.4±0.7 | 77.9±0.4 | 66.4±0.4 | 91.5±0.5 | 82.8±0.5 | 57.2±0.2 | 50.9±0.3 | ↑2.75% |
| OptBG+EDEN | 82.8±0.6 | 64.6±0.8 | 79.4±0.3 | 67.9±0.4 | 93.9±0.6 | 84.9±0.6 | 59.8±0.3 | 52.8±0.4 | |
| NAG | 81.2±0.9 | 62.5±0.9 | 78.3±0.3 | 65.9±0.5 | 91.3±0.7 | 83.1±0.4 | 57.1±0.2 | 51.2±0.4 | ↑2.54% |
| NAG+EDEN | 83.0±0.9 | 64.8±0.7 | 79.8±0.4 | 67.3±0.4 | 93.6±0.8 | 85.2±0.5 | 59.2±0.2 | 52.5±0.4 | |
| DIMPA | 82.4±0.6 | 64.0±0.8 | 78.8±0.4 | 67.1±0.3 | 91.4±0.6 | 82.4±0.5 | 56.7±0.3 | 50.5±0.3 | ⇑4.32% |
| DIMPA+EDEN | 85.4±0.5 | 66.9±0.7 | 82.2±0.5 | 69.9±0.3 | 94.1±0.7 | 85.1±0.5 | 59.5±0.4 | 52.9±0.2 | |
| Dir-GNN | 82.6±0.6 | 64.5±0.6 | 79.1±0.4 | 66.9±0.4 | 91.1±0.5 | 82.9±0.6 | 56.8±0.3 | 50.8±0.4 | ⇑4.68% |
| Dir-GNN+EDEN | 85.9±0.4 | 67.2±0.5 | 82.8±0.3 | 70.5±0.3 | 93.8±0.5 | 84.8±0.7 | 59.4±0.3 | 53.1±0.3 | |
| HoloNet | 82.5±0.5 | 64.1±0.7 | 79.2±0.3 | 67.5±0.2 | 90.8±0.5 | 83.0±0.6 | 57.0±0.3 | 51.0±0.4 | ⇑4.46% |
| HoloNet+EDEN | 86.0±0.4 | 67.5±0.6 | 82.6±0.2 | 70.8±0.3 | 93.7±0.5 | 85.3±0.5 | 59.5±0.5 | 53.4±0.5 | |

## 4.2 ABLATION STUDY

To answer **Q3**, we present ablation study results in Table 4, evaluating the effectiveness of the following modules: (1) Diverse knowledge in Eq. (9) for over-fitting issues; (2) Node-adaptive personalized transfer for KD (Eq. (10)); (3) Tree-based random walk for leaf prediction (Eq. (11)); (4) KD loss function for the gradient interaction between teachers and students (Eq. (10)). Notably, HKT serves as the core of the proposed EDEN framework, with the graph data online KD occurring within the layers of this tree structure, as shown in Fig. 2. As the foundational component of EDEN, the framework cannot function without this tree structure. Therefore, analyzing HKT in isolation during ablation studies is inappropriate. Instead, we highlight the contributions of each module designed to enable the effective implementation of EDEN, as detailed below.

Experimental results demonstrate a significant improvement in model predictions and variance reduction by combining these modules, validating their effectiveness. Specifically, module (1) introduces diverse knowledge from other partitions, mitigating over-fitting issues caused by solely focusing on the current partition. This effectiveness is reflected in higher accuracy and lower variance. Module (2) affirms our key insight: the need for a tailored knowledge transfer strategy for parent and child nodes in HKT-based KD. Thus, we can leverage KD to provide more effective supervision during model train-

Table 4: Ablation study performance (%).

| Datasets($\rightarrow$) | Tolokers (ACC) | | Slashdot (AUC) | |
|---|---|---|---|---|
| Modules($\downarrow$) | Node-C | Link-C | Existence | Direction |
| EDEN | **81.33±0.2** | **82.67±0.1** | **91.82±0.1** | **93.29±0.1** |
| w/o Diverse Knowledge | 81.10±0.3 | 82.32±0.2 | 91.50±0.2 | 93.06±0.2 |
| w/o Personalized Transfer | 81.04±0.2 | 82.44±0.1 | 91.59±0.1 | 93.11±0.1 |
| w/o Tree-based Random Walk | 80.87±0.3 | 82.28±0.2 | 91.26±0.1 | 92.87±0.1 |
| w/o Knowledge Distillation Loss | 80.21±0.3 | 81.20±0.1 | 90.94±0.1 | 92.35±0.1 |

| Datasets($\rightarrow$) | Rating (ACC) | | Epinions (AUC) | |
|---|---|---|---|---|
| Modules($\downarrow$) | Node-C | Link-C | Existence | Direction |
| EDEN | **46.33±0.4** | **66.37±0.4** | **93.48±0.1** | **89.40±0.1** |
| w/o Diverse Knowledge | 45.96±0.5 | 66.10±0.5 | 93.21±0.2 | 89.12±0.1 |
| w/o Personalized Transfer | 46.12±0.3 | 66.04±0.3 | 93.15±0.1 | 89.14±0.1 |
| w/o Tree-based Random Walk | 46.04±0.4 | 65.82±0.5 | 93.12±0.1 | 89.09±0.1 |
| w/o Knowledge Distillation Loss | 45.65±0.5 | 65.19±0.4 | 92.67±0.1 | 88.71±0.1 |

ing. Module (3) indirectly underscores the validity of the EDEN, as the multi-level representations embedded in the HKT provide beneficial information for various downstream tasks. This introduces a richer HKT semantic context, leading to a significant improvement in prediction accuracy. Finally, module (4) unifies the above modules into an end-to-end optimization framework to empower the digraph learning process. Module (2) can be seen as a more detailed exploration of this component.

### 4.3 Efficiency Comparison

To answer **Q4**, we present the running efficiency report in Fig. 3, where EDEN is primarily divided into two segments: (1) The pre-processing step depicted in Fig. 3(a) showcases coarse-grained HKT construction, with the x-axis representing predefined tree height $h$; (2) The end-to-end training step depicted in Fig. 3(b). The x-axis denotes the selection of tree height $h$ and sampling coefficient $\kappa$ introduced by Sec. 3.1

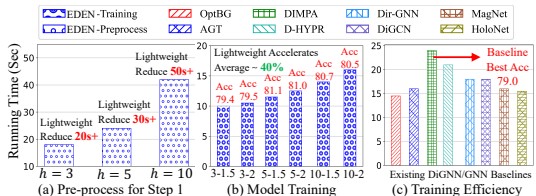

Figure 3: Efficiency of Node-C on Empire.

and Sec. 3.2. Since the pre-processing is independent of model training, the computational bottleneck introduced by the coarse-grained HKT construction is alleviated, reducing constraints on deployment scalability. Additionally, the lightweight implementation in pre-processing further mitigates it. Meanwhile, benefiting from the lightweight fine-grained HKT construction and personalized layer-wise digraph learning function, EDEN exhibits a significant advantage in training costs compared to existing baselines shown in Fig. 3(b)-(c). Due to space constraints, additional details regarding the model convergence efficiency during the training process can be found in Appendix A.14.

### 4.4 Robustness Analysis

**Hyperparameter Selection**. To answer **Q5**, we first analyze the impact of hyperparameter selection on running efficiency and predictive performance based on Fig. 3(a) and (b). Our observations include: (1) Higher HKT height $h$ leads to a substantial increase in the time complexity for greedy algorithm during pre-process; (2) Larger sampling coefficients $\kappa$ indicate additional computational costs due to considering more nodes in the knowledge generation, especially pronounced with increased height $h$; (3) Appropriately increasing $h$ and $\kappa$ for fine-grained distillation significantly improves performance. However, excessive increase leads to apparent optimization bottlenecks, resulting in sub-optimal performance. In addition, we further discuss the implementation details of HKT-based random walk for leaf prediction and KD loss factor in Appendix A.14. This involves investigating the impact of transition probabilities between distinct identity nodes (i.e., parent, sibling, and children) during the sequence acquisition on predictive performance and further analyzing the effectiveness of KD.

**Sparsity Challenges**. Subsequently, we provide sparse experimental results in Fig. 4. For stimulating feature sparsity, we assume that the feature of unlabeled nodes is partially missing. In this case, methods that rely on the quantity of node representations like D-HYPR and NAG are severely compromised. Conversely, DiGCN and MGC exhibit robustness, as high-order propa-

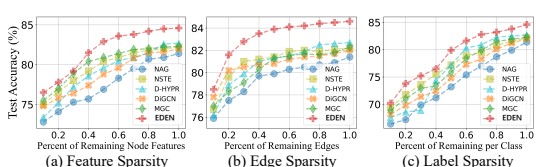

Figure 4: Node-C performance on CoraML.

gation partially compensates for missing features. As for edge sparsity, since all baselines rely on the topology to empower their neural architectures, their performance is not optimistic. However, we observe that EDEN exhibits leading performance through fine-grained digraph data knowledge mining. For stimulating label sparsity, we change the number of labeled samples for each class and acquire the results with a similar trend as the feature-sparsity tests. Building upon these observations, EDEN comprehensively improves both the performance and robustness of the various baselines.

## 5 Conclusions, Limitations, and Future Work

In this paper, we propose a general data-centric (di)graph online KD framework, EDEN. It achieves fine-grained data knowledge exploration abiding with the hierarchical thesis proposed in Sec. 2.2. Comprehensive evaluations demonstrate significant all-around advantages. We believe that implementing data-centric graph KD through the tree structure is a promising direction, as the hierarchical structure effectively captures the natural evolution of graphs. However, it must be acknowledged that the current EDEN framework has significant algorithmic complexity, including multi-step computations. Despite the lightweight implementation, scalability challenges persist when applied to billion-level graphs. Therefore, our future work aims to simplify the hierarchical data KD theory and develop a user-friendly computational paradigm to facilitate its practical deployment in industry.

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

# A APPENDIX

The appendix is organized as follows:

## A.1 HKT CONSTRUCTION AND HIERARCHICAL GRAPH CLUSTERING

Although the HKT construction process may appear similar to hierarchical clustering, it is important to clarify that HKT is fundamentally different, as it leverages topology-driven structural entropy, a dynamic measurement rooted in the information theory of structured data, to capture deeper structural insights. This approach goes beyond static clustering techniques, providing a more nuanced understanding of the underlying graph structure. Additionally, EDEN integrates feature-oriented node mutual information (MI) neural estimation as a key criterion for HKT construction, which allows for a more fine-grained analysis of node relationships based on feature information. As a result, the multi-granularity quantification criteria established by our method are not only distinct but also innovative compared to traditional hierarchical clustering (see Sec. 3.1).

While traditional hierarchical clustering can reveal the layered structure of a network, it is not directly applicable to the complexities of (di)graph learning. EDEN, on the other hand, utilizes HKT as a foundational framework to enable the development of learnable knowledge generation and transfer mechanisms that can be seamlessly integrated with existing (Di)GNN architectures. This integration provides a novel way to enhance model learning by effectively capturing and utilizing the hierarchical structure of directed graphs. Furthermore, we have designed a random walk-based leaf prediction mechanism, tailored to various graph-based downstream tasks, ensuring that our approach is robust and adaptable to different application scenarios (for more technical details, refer to Sec. 3.2-3.3).

## A.2 DATA-LEVEL ONLINE KNOWLEDGE DISTILLATION IN EDEN

Graph KD typically follows a model-level, offline teacher-student framework. In this setup, knowledge is transferred from a large, pre-trained teacher GNN to a more compact and efficient student model, such as a smaller GNN or MLP. The teacher captures complex patterns and representations within the graph. The student, rather than learning directly from ground truth labels, learns from the teacher's soft predictions or intermediate representations. This approach allows the student model to replicate the teacher's performance while significantly reducing computational complexity.

With the rapid advancement of KD, it has expanded into multiple model-level KD variants. These include self-distillation, where a single model simultaneously acts as both the teacher and student, enhancing its own learning process Chen et al. (2021); Zhang et al. (2023), and online distillation, where both teacher and student models are continuously updated throughout the training process Zhang et al. (2021b); Feng et al. (2022). These innovations reflect the growing diversity in how knowledge transfer can be applied beyond the initial teacher-student (large model to lightweight model) framework.

In this paper, we focus specifically on data-level graph KD, which emphasizes uncovering the latent knowledge embedded in graph structures, using data samples as the medium for distillation Zhang et al. (2020); Zhu et al. (2024). In the EDEN framework, parent and child nodes within the HKT assume the roles of teacher and student, respectively. This enables knowledge transfer through their representations in a hierarchical manner. Our approach aligns with the principles of data-level online KD, leveraging the topological relationships between nodes to drive more effective distillation.

### A.3 THE DEFINITION OF PARTITION TREE

To define high-dimensional measurements of directed structural information, we introduce a partition tree $\mathcal{T}$ of digraphs, which can also be regarded as the coarse-grained HKT without feature-oriented refinement (i.e., knowledge discovery (a) from a topology perspective only). Notably, community detection or clustering can be understood as a hierarchical structure, specifically a 3-layer partition tree. In this structure, the leaf nodes represent the individual nodes from the original graph, while their parent nodes serve as virtual nodes that represent entire communities. To make it easier to understand, we first give an example of a two-dimensional directed structural measurement of the graph, $\mathcal{H}^2(\mathcal{G})$, where we consider a digraph $\mathcal{G} = (\mathcal{V}, \mathcal{E})$ and its 2-order partition $\mathcal{P} = \{\mathcal{X}_1, \cdots, \mathcal{X}_{\mathcal{C}}\}$ of node sets $\mathcal{V}$. Building upon this, we interpret $\mathcal{P}$ through a 2-height partition tree $\mathcal{T}$ as follows.

To begin with, we introduce the root node $\lambda$ and define a set of nodes $T_\lambda = \mathcal{V}$ as a subset of the root node $\lambda$ in the 2-height partition tree $\mathcal{T}$. Notably, in this two-dimensional directed structural measurement, the nodes in the 2-height partition tree have only three types of identity information:

(1) the *root* node ($h = 0$), which does not exist in the original digraph but is used to describe the partition tree;

(2) the *successor* nodes ($h = 1$), which are not present in the original digraph but are employed to characterize leaf nodes;

(3) the *leaf* nodes ($h = 2$), which represent the original digraph nodes.

Then, we introduce $\mathcal{C}$ immediate successors for the root denote $\phi_i = \lambda\langle i\rangle$, $i = 1, \cdots, \mathcal{C}$. Naturally, we can extend the concept associated with the root to successor nodes $\phi_i$, which are directly related to the coarse partitioning of leaf nodes $\mathcal{X}_i$. Thus, we define $T_{\phi_i} = \mathcal{X}_i$. Now, for each $\phi_i$, we introduce $|\mathcal{X}_i|$ immediate successors denoted $\phi_i\langle j\rangle$ for all $j \in \{1, \cdots, |\mathcal{X}_i|\}$, and each successor $\phi_i\langle j\rangle$ is associated with an element in $\mathcal{X}_i$. Thus, we define $T_{\phi_i\langle j\rangle}$ as the singleton of a node in $T_{\phi_i} = \mathcal{X}_i$.

To this point, $\mathcal{T}$ is a partition tree of height 2, and all its leaves are associated with singletons. For any node $\alpha \in \mathcal{T}$, $T_\alpha$ is the union of $T_\beta$ for all $\beta$ values (immediate successors) of $\alpha$, and the union of $T_\alpha$ for all nodes with $\alpha$ values at the same level of the partition tree $\mathcal{T}$ constitutes a partition of $\mathcal{V}$. Hence, the partition tree of a digraph is a set of nodes, each associated with a nonempty subset of nodes in digraph $\mathcal{G}$, and can be defined as follows:

**Definition A.1.** (partition tree of Digraphs): Let $\mathcal{G} = (\mathcal{V}, \mathcal{E})$ be a connected digraph. We define the $h$-height partition tree $\mathcal{T}$ of $\mathcal{G}$ with the following properties:

(1) For the root node $\lambda$, we define the set $T_\lambda = \mathcal{V}$ as the collection of nodes with heights less than $\lambda$.

(2) For each node $\alpha \in \mathcal{T}$, the immediate successors of $\alpha$ are denoted as $\alpha\langle j\rangle$ for $j$ ranging from 1 to a natural number $N$, ordered from left to right as $j$ increases.

(3) For any natural number $i \leq h$ and each non-leaf node $\alpha \neq \lambda$, the set $\{T_\alpha \mid h(\alpha) = i\}$ forms a partition of $\mathcal{V}$, where $h(\alpha)$ denotes the height of $\alpha$ (note that the height of the root node $\lambda$ is 0).

(4) For each leaf node $\alpha$ in $\mathcal{T}$, $T_\alpha$ is a singleton, indicating that $T_\alpha$ contains a single node from $\mathcal{V}$.

(5) For any two nodes $\alpha, \beta \in \mathcal{T}$ at different heights in the tree, we use $\alpha \subset \beta$ or $\beta \subset \alpha$ to denote their hierarchical relationship.

(6) For $\alpha \subset \beta$ or $\beta \subset \alpha$, we employ $-$ and $+$ to further describe this hierarchical relationship within the same partition. Specifically, if $\alpha \subset \beta$ with $h(\alpha) = h(\beta) + 1$, then $\beta^-$ represents the child nodes of $\beta$. Conversely, if $\beta \subset \alpha$ with $h(\beta) = h(\alpha) + 1$, then $\beta^+$ denotes the parent node of $\beta$. (note that for every non-leaf node $\alpha \neq \lambda$, $h(\alpha^-) - 1 = h(\alpha) = h(\alpha^+) + 1$)

(7) For each $\alpha$, $T_\alpha$ is the union of $T_\beta$ for all $\beta$ such that $\beta^+ = \alpha$. Thus, $T_\alpha = \cup_{\beta^+=\alpha} T_\beta$.

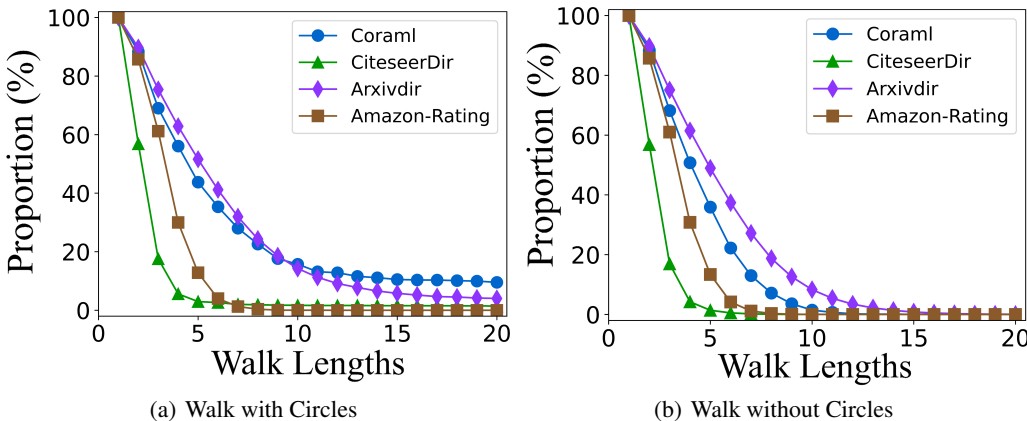

(a) Walk with Circles            (b) Walk without Circles

Figure 5: The visualization experiments of the interruption issue of single-direction random walk on digraphs. The circle represents a special topology where a node can walk back to itself, and its existence will alleviate walk interruption. The y-axis denotes the proportion of non-walk interruptions.

According to Definition A.1, for a given digraph $\mathcal{G}$, we compute the $h$-dimensional directed structural information measurement $\mathcal{H}^h(\mathcal{G})$ of $\mathcal{G}$ by Eq. (3) while simultaneously identifying a $h$-height coarse-grained HKT $\mathcal{T}$. The above process adheres to the following principles:

(1) The $h$-dimensional structural information measurement $\mathcal{H}^h(\mathcal{G})$ of a digraph $\mathcal{G}$ is achieved or approximated through the $h$-dimensional hierarchical partition tree $\mathcal{T}$ of $\mathcal{G}$;

(2) $\mathcal{H}^h(\mathcal{G})$ serves as the guiding principle for the formation of the $h$-dimensional coarse-grained HKT $\mathcal{T}$ by minimizing the uncertainty or non-determinism inherent in the $h$-dimensional structures of $\mathcal{G}$;

(3) $\mathcal{T}$, functioning as a coarse-grained HKT for $\mathcal{G}$, encompasses the rules, regulations, and orders governing $\mathcal{G}$. This HKT is derived by minimizing the random variations present in the $h$-dimensional structures of the digraphs, with these variations being determined by our $h$-dimensional directed structural information measurement.

Based on the above principles, the $h$-dimensional structural measurement of digraphs, provided by the $h$-height partition tree, serves as a metric enabling us to comprehensively or maximally identify the $h$-dimensional structure while mitigating the impact of random variations in the digraphs. Meanwhile, $\mathcal{H}^h(\mathcal{G})$ excellently facilitates the complete extraction of order from unordered digraphs, allowing us to discern order from disorder within structured data. Remarkably, our definition retains all properties of the digraphs, providing robust support for the thorough analysis of structured data.

### A.4 BREAKING THE LIMITATIONS OF SINGLE-DIRECTION RANDOM WALKS

Utilizing simple random walks (SRW) on digraphs introduces unique challenges due to the inherent structure of these graphs. A common issue arises when the random walk encounters nodes with no outgoing edges, causing the walk to terminate prematurely. To better understand and visualize this limitation, we apply SRW starting from each node across four different digraphs. As the walk length increases, we track the proportion of complete paths relative to the total sequences, as shown in Fig. 5(a). To further assess the impact of graph cycles, we design a modified SRW that excludes cycles and conduct the same experiment, with results presented in Fig. 5(b).

This investigation highlights a key limitation of random walks on digraphs: strictly following edge directions leads to frequent interruptions in the walk. Due to the non-strongly connected nature of most digraphs, the proportion of complete walks drops sharply after just five steps. This indicates that random walks on digraphs typically fail to gather information beyond the immediate neighborhood of the starting node, limiting their ability to capture long-range dependencies. Moreover, when we eliminate the influence of cycles, the proportion of uninterrupted sequences declines even further, underscoring the difficulty of maintaining continuous paths in digraphs and further highlighting the limitations of SRWs (forward-only) in exploring deeper graph structures. It is evident that this significantly hinders the ability of structural entropy to capture topological uncertainty, reducing the effectiveness of $\mathcal{H}^h(\mathcal{G})$ and leading to sub-optimal coarse-grained HKT.

## A.5 GREEDY ALGORITHMS FOR PARTITION TREE CONSTRUCTION

The primary impetus for developing the greedy partition tree construction algorithm lies in the quest for an effective method to construct hierarchical tree structures from digraph data while simultaneously minimizing the complexity and uncertainty associated with the underlying relationships. In complex systems represented by digraphs, directed structural entropy serves as a key metric to gauge the disorder and intricacy within the network. By harnessing the concept of directed edge structural entropy minimization, the algorithm aims to derive hierarchical trees that capture essential structural characteristics while promoting simplicity and interpretability. In a nutshell, the design principles of our proposed algorithm are as follows

(1) Directed edge structural entropy definition: The algorithm hinges on a rigorous definition of directed edge structural entropy within the context of the digraph mentioned in Sec. 3.1. This metric quantifies the uncertainty and disorder associated with the relationships between nodes in the digraph.

(2) Greedy selection strategy: At its core, the algorithm employs a greedy strategy, iteratively selecting directed edges that contribute most significantly to the reduction of directed structural entropy. This strategy ensures that each step in the tree construction process maximally minimizes the overall disorder in the evolving hierarchy.

(3) Hierarchical tree construction: The selected directed edges are systematically incorporated into the growing tree structure, establishing a hierarchical order that reflects the inherent organization within the graph. This process continues iterations until a coherent and informative tree representation is achieved.

(4) Complexity considerations: The algorithm balances the trade-off between capturing essential structural information and maintaining simplicity. By prioritizing directed edges that significantly impact entropy reduction, it aims to construct trees that are both insightful and comprehensible.

In conclusion, the greedy partition tree construction algorithm for digraph data, rooted in the minimization of directed edge structural entropy, presents a promising avenue for extracting hierarchical structures from the network with intricate topology. To clearly define a greedy partition tree construction algorithm, we introduce the following meta-operations in Alg. 1.

These meta-operations collectively define the intricate logic underlying the greedy partition tree construction algorithm, providing a comprehensive framework for constructing hierarchical structures in graph data while adhering to the principles of minimizing directed edge structural entropy. Building upon these foundations, we employ meta-operations to present the detailed workflow of the greedy structural tree construction algorithm. This facilitates the coarse-grained HKT construction from a topological perspective, ultimately achieving digraph data knowledge discovery (i.e., Step 1 Knowledge Discovery (a) in our proposed EDEN as illustrated in Fig. 2).

The Alg. 2 outlines the construction of a height-limited partition tree algorithm, emphasizing the minimization of directed structural uncertainty. It begins by sorting input data in non-decreasing order. Subsequently, it constructs an initial partition tree, using a greedy approach that iteratively combines nodes until the root has only two children. After that, it enters a phase of height reduction, wherein nodes contributing to excess height are detached iteratively until the tree attains height $h$. To stabilize the structure, it inserts filler nodes for any node with a height discrepancy exceeding 1. This three-phase process ensures the efficient construction of a height-limited partition tree while minimizing directed structural measurement.

## A.6 THE PROOF OF THEOREM 3.1

As discussed in Sec. 3.1, node features in a digraph act as essential identifiers that exhibit strong correlations with node labels. These features are not only instrumental in distinguishing nodes but also play a critical role in the construction of data knowledge. Recognizing this, our proposed partition-based node MI neural estimation seeks to further refine the coarse-grained HKT, which is initialized by the greedy algorithm. This refinement is achieved by quantifying the correlations between node features within the partition tree, thereby enhancing the granularity of the HKT. The refined tree provides a more accurate and nuanced representation of the graph, laying a robust foundation for subsequent KD. This process ensures that both topological structure and node feature information are effectively leveraged in the distillation, leading to improved model performance.

---

**Algorithm 1** Meta-operation (Function) Definitions

---

**Definition.** node $v_\lambda$ is the root of $\mathcal{T}$, nodes $(v_i, v_j)$ are two children nodes of node $v_\lambda$

// **Meta-1**: Counts the number of children nodes of the given node $v_\lambda$.
**Function** CountChildren($v_\lambda$):
**Return** Number of children of node $\lambda$

// **Meta-2**: Inserts a new node between nodes $v_i$ and $v_j$, with $v_\lambda$ as the root.
**Function** Combine($v_i, v_j$):
Insert a new node $v_n$ between nodes $v_i$, $v_j$ and node $v_\lambda$
$v_\lambda.children \leftarrow v_n$
$v_n.children \leftarrow v_i$
$v_n.children \leftarrow v_j$

// **Meta-3**: Chooses two nodes $(v_i, v_j)$ from $v_\lambda.children$ to maximize the reduction of $\mathcal{H}^{\mathcal{T}}(\mathcal{G})$.
**Function** PickTwo($G$):
$\mathrm{argmax}_{(v_i, v_j)} \left\{ \mathcal{H}^T(G) - \mathcal{H}^{T_{\mathrm{Combine}(v_i, v_j)}}(G) \right\}$
**Return** $(v_i, v_j)$

// **Meta-4**: Computes the height of the partition tree $\mathcal{T}$.
**Function** TreeHeight($\mathcal{T}$):
**Return** $h(\mathcal{T})$

// **Meta-5**: Detaches node $v_i$ from the tree $\mathcal{T}$ and merges its children to $v_j.children$.
**Function** Detach($v_i$):
Detach $v_i$ from $\mathcal{T}$ and merge its children to $v_j.children$
$v_j.children \leftarrow v_j.children + v_i.children$
**Delete** $v_i$

// **Meta-6**: Chooses one node $v_i$ from $\mathcal{T}$ based on minimizing the increase of $\mathcal{H}^{\mathcal{T}}(\mathcal{G})$.
**Function** ChooseNode($\mathcal{T}$):
$\mathrm{argmin}_{v_i} \left\{ \mathcal{H}^{\mathcal{T}_{\mathrm{detach}(v_i)}}(\mathcal{G}) - \mathcal{H}^{\mathcal{T}}(\mathcal{G}) \mid v_i \neq v_r \right\}$
**Return** $v_i$

// **Meta-7**: Computes the absolute difference in height between the parent of $v_i$ and $v_i$.
**Function** DeltaHeight($v_i$):
**Return** $\mid \mathrm{TreeHeight}(v_i.parent) - \mathrm{TreeHeight}(v_i) \mid$

// **Meta-8**: Inserts a filler node between nodes $v_i$ and $v_j$ to keep the tree height balanced.
**Function** InsertFillerNode($v_i, v_j$):
Insert a new node $v_n$ between nodes $v_i$ and $v_j$
$v_n.children \leftarrow v_i$
$v_j.children \leftarrow v_n$

---

---

**Algorithm 2** Construction of a Height-Limited Partition Tree

---

**Input:** data $x_i$, size $m$
**repeat**
  Initialize $noChange$ = true.
  **for** $i = 1$ **to** $m - 1$ **do**
    **if** $x_i > x_{i+1}$ **then**
      Swap $x_i$ and $x_{i+1}$
      $noChange$ = false
    **end if**
  **end for**
**until** $noChange$ is true
**Input:** a digraph $\mathcal{G} = (\mathcal{V}, \mathcal{E})$, an integer $h \geq 2$
Initialize partition tree $\mathcal{T}$ with root node $\lambda$ and set all $\mathcal{V}$ as leaves

Phase I: Build a partition tree from leaves to root, using the greedy method
**while** CountChildren($r$) > 2 **do**
  $(v_i, v_j) \leftarrow$ PickTwo($\mathcal{G}$)
  Combine($v_i, v_j$) $\rightarrow \mathcal{T}$
**end while**

Phase II: Height reduction to $h$
**while** TreeHeight($\mathcal{T}, \mathcal{G}, \mathcal{V}'$) > $h$ **do**
  $v_i \leftarrow$ ChooseNode($\mathcal{T}, \mathcal{G}, \mathcal{V}'$)
  Detach($v_i$) from $\mathcal{T}$
**end while**

Phase III: Stabilize tree structure
**for** Each $v_i \in \mathcal{T}$ **do**
  **if** DeltaHeight($v_i$) > 1 **then**
    InsertFillerNode($v_i, v_i$.parent)
  **end if**
**end for**

---

Considering a digraph $\mathcal{G} = (\mathcal{V}, \mathcal{E})$ and its coarse-grained partition tree $\mathcal{T}$, where $\mathcal{V}$ encompasses all nodes in the digraph, along with the corresponding feature and label matrix represented as $\mathbf{X}$ and $\mathbf{Y}$. For current partition $\mathcal{X}_p$ given by $\mathcal{T}$, we employ a sampling strategy to obtain a candidate node subset $\Omega_p$ with $K_p$ nodes from the current partition $\mathcal{X}_p$ and other partitions $\mathcal{X}_q$. Notably, different partitions used for sampling should be at the same height within the HKT (e.g., the current partition $\mathcal{X}_p$ and other partitions $\mathcal{X}_q$ should satisfy $h(\mathcal{X}_p) = h(\mathcal{X}_q)$). Building upon this, to reduce the computational complexity, we adopt a computation-friendly sampling strategy. Specifically, considering the number of nodes in the current partition is $|\mathcal{X}_p|$, we include all of them in the candidate set $\Omega_p$. Additionally, we perform random sampling for partition-by-partition until the total non-duplicated nodes in the $\Omega_p$ satisfy $\kappa |\mathcal{X}_p|$, where $\kappa \geq 1$ is used to control the knowledge domain expansion come from the other partitions $\mathcal{X}_q$. This subset $\Omega_p$ is used to generate knowledge that represents the current partition $\mathcal{X}_p$, formally represented as the parent representation of this partition in the HKT. Notably, we assign distinct identifiers to the sampled nodes based on their partition affiliations, denoting them as $v \in \mathcal{X}_p$ and $u \in \mathcal{X}_q$, providing clarity in illustrating our method and derivation process.

Building upon this foundation, given the node $v$ as an example, a random variable $f_v$ is introduced to represent the node feature when randomly selecting a node from $\Omega_p$ within the current partition $\mathcal{X}_p$. Then, the probability distribution of $f_v$ is formally defined as $P_{f_v} = P(f_v = \mathbf{X}_v), \forall v \in \Omega_p \cap \mathcal{X}_p$. Similarly, we can generalize $P_{f_v}$ to scenarios originating from other partitions to obtain $P_{f_u} = P(f_u = \mathbf{X}_u), \forall u \in \Omega_p \cap \mathcal{X}_q$. In $\Omega_p$, the definition of the generalized neighborhoods for any node is closely tied to the partition provided by the HKT, rather than relying on the traditional definition based on the adjacency matrix $\mathbf{A}$ from directed edge sets $\mathcal{E}$. Specifically, for nodes belonging to the current partition, denoted as $v \in \mathcal{X}_p$, their generalized neighborhoods are defined as $\mathcal{N}_v^{\mathcal{T}} = \mathcal{X}_p$. This is done to identify nodes with sufficient information to efficiently represent the current partition i.e., (measure MI between $v$ and $\mathcal{N}_v^{\mathcal{T}}$). As for nodes belonging to other partitions, denoted as $u \in \mathcal{X}_q$, their

generalized neighborhoods are defined as $\mathcal{N}_u^{\mathcal{T}} = \mathcal{X}_p \cup \mathcal{X}_q$. This is intended to address the limitations of the coarse-grained partition tree produced by considering only topological metrics. In other words, we aim to identify sets of nodes within other partitions that effectively capture the representation of both the current partition $\mathcal{X}_p$ (explore potential correlation from the feature perspective) and their own partition $\mathcal{X}_q$ (inherit their own partition criteria about directed structural information measurement), thereby refining the HKT through MI measurement between $u$ and $\mathcal{N}_u^{\mathcal{T}}$.

Notably, we chose $\mathcal{N}_v^{\mathcal{T}} = \mathcal{X}_p$ for the following reasons: (1) We aim to calculate the MI neural estimation between the current node $v$ and its generalized neighborhoods $\mathcal{X}_p$ as a criterion for quantifying affinity scores. This approach ensures that nodes representative of the current partition receive higher affinity scores. Therefore, the generalized neighborhood of the current node needs to be closely related to the partition to which the node belongs, leading us to impose this restriction rather than defining the neighborhood as all nodes $\mathcal{V}$. For more on the motivation, intuition, and theory behind this mechanism, please refer to Sec. 2.2. As for the details on the calculation of affinity scores, we recommend referring to Sec. 3.2 on knowledge generation. (2) In general, the number of partitions $\mathcal{X}_p$ is considerably smaller than the total set of nodes $\mathcal{V}$. As a result, one of the key motivations for imposing this neighborhood restriction is to minimize computational overhead and improve overall runtime efficiency. By limiting the scope of the calculations, we are able to streamline the process without sacrificing performance, making the method more scalable for large-scale graphs. In summary, expanding the neighborhood to include all nodes would result in higher computational costs and poorer performance. Therefore, we restrict the definition of the generalized neighborhood based on the partition obtained by HKT.

In either case, the generalized neighborhoods are subgraphs containing nodes from $\mathcal{V}$. These nodes may not be directly connected in the original topology but reveal inherent correlations at a higher level through the measurement of directed structural information. Therefore, this representation transcends the topological exploration of the digraph by $\mathbf{A}$ and reflects intrinsic knowledge at a higher level. Building upon this, considering a node $v$ as an example, let $f_{\mathcal{N}_v^{\mathcal{T}}}$ be a random variable representing the generalized neighborhood feature selected from $\Omega_p$, originating from the current partition $\mathcal{X}_p$. We define the probability distribution of $f_{\mathcal{N}_v^{\mathcal{T}}}$ as $P_{f_{\mathcal{N}_v^{\mathcal{T}}}} = P(f_{\mathcal{N}_v^{\mathcal{T}}} = \mathbf{X}_{\mathcal{N}_v^{\mathcal{T}}})$.

Therefore, considering a node $v \in \mathcal{X}_p$ as an example, we define the joint distribution of the random variables of node features and its generalized neighborhood features within partition $\mathcal{X}_p$ given by HKT, which is formulated as:

$$P\left(f_v, f_{\mathcal{N}_v^{\mathcal{T}}}\right) = P\left(f_v = \mathbf{X}_v, f_{\mathcal{N}_v^{\mathcal{T}}} = \mathbf{X}_{\mathcal{N}_v^{\mathcal{T}}}\right), \forall v \in \Omega_p \cap \mathcal{X}_p, \tag{13}$$

where the joint distribution reflects the probability that we randomly pick the corresponding node feature and its generalized neighborhood feature of the same node $v$ within partition $\mathcal{X}_p$ together. Building upon this, the MI between the node features and the generalized neighborhood features within the current partition $\mathcal{X}_p$ is defined as the KL-divergence between the joint distribution $P\left(f_v, f_{\mathcal{N}_v^{\mathcal{T}}}\right)$ and the product of the marginal distributions of the two random variables $P_{f_v} \otimes P_{f_{\mathcal{N}_v^{\mathcal{T}}}}$. The above process can be formally defined as:

$$\mathcal{I}^{(\Omega)}\left(f_v, f_{\mathcal{N}_v^{\mathcal{T}}}\right) = \mathcal{D}_{\mathrm{KL}}\left(P\left(f_v, f_{\mathcal{N}_v^{\mathcal{T}}}\right) \| P_{f_v} \otimes P_{f_{\mathcal{N}_v^{\mathcal{T}}}}\right). \tag{14}$$

This MI measures the mutual dependency between the selected node and its generalized neighborhoods in $\Omega_p$. The KL divergence adopts the $f$-representation Belghazi et al. (2018) is defined as:

$$\begin{aligned}
\mathcal{D}_{\mathrm{KL}}\left(P\left(f_v, f_{\mathcal{N}_v^{\mathcal{T}}}\right) \| P_{f_v} \otimes P_{f_{\mathcal{N}_v^{\mathcal{T}}}}\right) \geq & \sup_{F \in \mathcal{F}}\left\{\mathbb{E}_{\mathbf{X}_v, \mathbf{X}_{\mathcal{N}_v^{\mathcal{T}}} \sim P\left(f_v, f_{\mathcal{N}_v^{\mathcal{T}}}\right)}\left[F\left(\mathbf{X}_v, \mathbf{X}_{\mathcal{N}_v^{\mathcal{T}}}\right)\right]\right\} \\
& - \sup_{F \in \mathcal{F}}\left\{\mathbb{E}_{\mathbf{X}_v \sim P_{f_v}, \mathbf{X}_{\mathcal{N}_v^{\mathcal{T}}} \sim P_{f_{\mathcal{N}_v^{\mathcal{T}}}}}\left[e^{F\left(\mathbf{X}_v, \mathbf{X}_{\mathcal{N}_v^{\mathcal{T}}}\right)-1}\right]\right\},
\end{aligned} \tag{15}$$

where $\mathcal{F}$ is an arbitrary class of functions that maps a pair of selected node features and its generalized neighborhood features to a real value. Here, we use $F(\cdot, \cdot)$ to compute the dependency. If we explore any possible function $F \in \mathcal{F}$, it can serve as a tight lower bound for MI. Building upon this, we can naturally extend the above derivation process to the scenario of sampling nodes belonging to other partitions, specifically $u \in \Omega_p \cap \mathcal{X}_q$. At this point, we can assess the shared contribution of nodes $v$ and $u$ with different affiliations in generating knowledge for the current partition $\mathcal{X}_p$.

## A.7   THE PROOF OF THEOREM 3.2

The primary objective here is to introduce a node selection criterion that is grounded in quantifying the dependency between the selected node and its generalized neighborhoods. This dependency serves as the foundation for assessing the relevance and influence of each node within its local structure. The key insights behind using this dependency as a guiding principle are central to the formulation of the criterion function. By leveraging this approach, we aim to enhance the process of knowledge generation for the current partition $\mathcal{X}_p$, ensuring that both local and global relationships are effectively captured and utilized in the knowledge distillation process. The detailed reasoning and benefits of this approach are outlined as follows:

(1) In our definition, the generalized neighborhoods of the selected node are closely tied to the current partition $\mathcal{X}_p$ and their own partition $\mathcal{X}_i$. Thus, measuring this dependency is equivalent to quantifying the correlation between the representation of the selected node and the knowledge possessed by the current partition and their own partition.

(2) The node-selection criterion is essentially a mechanism for weight allocation. Since the candidate node set is fixed by the sampling process, this step aims to assign higher affinity scores to nodes that better represent the current and their own partition. This guides the knowledge generation process to acquire the parent node representation for the current partition.

Building upon this, instead of calculating the exact MI based on KL divergence, we opt for non-KL divergences to offer favorable flexibility and optimization convenience. Remarkably, both non-KL and KL divergences can be formulated within the same $f$-representation framework. We commence with the general $f$-divergence between the joint distribution and the product of marginal distributions of vertices and neighborhoods. The above process can be formally defined as follows:

$$
\mathcal{D}_f \left( P \left( f_v, f_{\mathcal{N}_v^{\mathcal{T}}} \right) \| P_{f_v} \otimes P_{f_{\mathcal{N}_v^{\mathcal{T}}}} \right) = \int P_{f_v} P_{f_{\mathcal{N}_v^{\mathcal{T}}}} f \left( \frac{P \left( f_v, f_{\mathcal{N}_v^{\mathcal{T}}} \right)}{P_{f_v} P_{f_{\mathcal{N}_v^{\mathcal{T}}}}} \right) d\mathbf{X}_v d\mathbf{X}_{\mathcal{N}_v^{\mathcal{T}}}, \quad (16)
$$

where $f(\cdot)$ represents a convex and lower-semicontinuous divergence function. When $f(x) = x \log x$, the $f$-divergence is specified as the Kullback-Leibler (KL) divergence. The function $f(\cdot)$ has a convex conjugate function, denoted as $f^\star(\cdot)$, where $f^\star(t) = \sup_{x \in \text{dom}_f} \{tx - f(x)\}$, and $\text{dom}_f$ is the domain of $f(\cdot)$. It's important to note that these two functions, $f(\cdot)$ and $f^\star(\cdot)$, are dual to each other. According to the Fenchel conjugate Hiriart-Urruty & Lemaréchal (2004) and node sampling space $\Omega_p$ based on different affiliations given by HKT, the $f$-divergence can be modified as:

$$
\begin{aligned}
&\mathcal{D}_f \left( P \left( f_v, f_{\mathcal{N}_v^{\mathcal{T}}} \right) \| P_{f_v} \otimes P_{f_{\mathcal{N}_v^{\mathcal{T}}}} \right) \\
&= \int P_\mathbf{X} P_{f_{\mathcal{N}_v^{\mathcal{T}}}} \sup_{t \in \text{dom}_{f^\star}} \left\{ t \frac{P \left( \mathbf{X}, f_{\mathcal{N}_v^{\mathcal{T}}} \right)}{P_{f_v} P_{f_{\mathcal{N}_v^{\mathcal{T}}}}} - f^\star(t) \right\} \\
&\geq \sup_{F \in \mathcal{F}} \left\{ \mathbb{E}_{P \left( f_v, f_{\mathcal{N}_v^{\mathcal{T}}} \right)} \left[ F \left( \mathbf{X}_v, \mathbf{X}_{\mathcal{N}_v^{\mathcal{T}}} \right) \right] - \mathbb{E}_{P_{f_v}, P_{f_{\mathcal{N}_v^{\mathcal{T}}}}} \left[ f^\star \left( F \left( \mathbf{X}_v, \mathbf{X}_{\mathcal{N}_{\bar{v}}^{\mathcal{T}}} \right) \right) \right] \right\},
\end{aligned} \quad (17)
$$

where $\mathcal{F}$ represents any function that maps the selected node and its generalized neighborhood features to a scalar, and the function $F(\cdot, \cdot)$ serves as a variational representation of $t$. $\bar{v}$ is a randomly selected node from $\Omega_p$ excluding $v$. This step confines the quantification of MI to the sampling space of $\Omega_p$, providing a finer-grained quantification criterion. Additionally, we employ an activation function $\sigma : \mathbb{R} \rightarrow \text{dom}_{f^\star}$ to constrain the function value $F(\cdot, \cdot) \rightarrow \sigma(F(\cdot, \cdot))$. Thus, we obtain:

$$
\begin{aligned}
&\mathcal{D}_f \left( P \left( f_v, f_{\mathcal{N}_v^{\mathcal{T}}} \right) \| P_{f_v} \otimes P_{f_{\mathcal{N}_v^{\mathcal{T}}}} \right) \geq \\
&\sup_{F \in \mathcal{F}} \left\{ \mathbb{E}_{P \left( f_v, f_{\mathcal{N}_v^{\mathcal{T}}} \right)} \left[ \sigma \left( F \left( \mathbf{X}_v, \mathbf{X}_{\mathcal{N}_v^{\mathcal{T}}} \right) \right) \right] - \mathbb{E}_{P_{f_v}, P_{f_{\mathcal{N}_v^{\mathcal{T}}}}} \left[ f^\star \left( \sigma \left( F \left( \mathbf{X}_v, \mathbf{X}_{\mathcal{N}_{\bar{v}}^{\mathcal{T}}} \right) \right) \right) \right] \right\}.
\end{aligned} \quad (18)
$$

Given that $\sigma(F(\cdot, \cdot))$ also belongs to $\mathcal{F}$ and its value falls within $\text{dom}_{f^\star}$, the optimal solution satisfies the equation. Assuming the divergence function is $f(x) = x \log x$, the conjugate divergence function is $f^\star(t) = \exp(t - 1)$, and the activation function is $\sigma(x) = x$, we can derive the $f$-representation of KL divergence shown in Eq. (15). It is important to note that the choice of the

activation function $\sigma(\cdot)$ is not unique, and our target is to identify one that facilitates both derivation and computation. Here, we explore an alternative form of divergence utilizing $f$-representation, known as GAN-like divergence. In this context, we employ a specific form of the divergence function, given by $f(x) = x \log x - (x+1) \log(x+1)$, with the conjugated divergence function defined as $f^\star(t) = -\log(1 - \exp(t))$ Nowozin et al. (2016). The chosen activation function is $\sigma(\cdot) = -\log(1 + \exp(\cdot))$. The GAN-like divergence can be expressed as:

$$
\mathcal{D}_{\mathrm{GAN}}\left(P\left(f_v, f_{\mathcal{N}_v^{\mathcal{T}}}\right) \| P_{f_v} \otimes P_{f_{\mathcal{N}_v^{\mathcal{T}}}}\right)
$$

$$
\geq \sup_{F \in \mathcal{F}} \left\{ \mathbb{E}_{P\left(f_v, f_{\mathcal{N}_v^{\mathcal{T}}}\right)} \left[\sigma\left(F\left(\mathbf{X}_v, \mathbf{X}_{\mathcal{N}_v^{\mathcal{T}}}\right)\right)\right] - \mathbb{E}_{P_{f_v}, P_{f_{\mathcal{N}_v^{\mathcal{T}}}}} \left[f^\star\left(\sigma\left(F\left(\mathbf{X}_v, \mathbf{X}_{\mathcal{N}_{\tilde{v}}^{\mathcal{T}}}\right)\right)\right)\right] \right\}
$$

$$
= \sup_{F \in \mathcal{F}} \left\{ \mathbb{E}_{P\left(f_v, f_{\mathcal{N}_v^{\mathcal{T}}}\right)} \left[-\log\left(1 + \exp\left(-\sigma\left(\mathbf{X}_v, \mathbf{X}_{\mathcal{N}_v^{\mathcal{T}}}\right)\right)\right)\right] \right\}
$$

$$
+ \sup_{F \in \mathcal{F}} \left\{ \mathbb{E}_{P_{f_v}, P_{f_{\mathcal{N}_v^{\mathcal{T}}}}} \log\left(1 - \exp\left(-\log\left(1 + e^{F\left(\mathbf{X}_v, \mathbf{X}_{\mathcal{N}_{\tilde{v}}^{\mathcal{T}}}\right)}\right)\right)\right) \right\}
$$

$$
= \sup_{F \in \mathcal{F}} \left\{ \mathbb{E}_{P\left(f_v, f_{\mathcal{N}_v^{\mathcal{T}}}\right)} \log \frac{1}{1 + e^{-F\left(\mathbf{X}_v, \mathbf{X}_{\mathcal{N}_v^{\mathcal{T}}}\right)}} \right\} \tag{19}
$$

$$
+ \sup_{F \in \mathcal{F}} \left\{ \mathbb{E}_{P_{f_v}, P_{f_{\mathcal{N}_v^{\mathcal{T}}}}} \log\left(1 - \frac{1}{1 + e^{-F\left(\mathbf{X}_v, \mathbf{X}_{\mathcal{N}_{\tilde{v}}^{\mathcal{T}}}\right)}}\right) \right\}
$$

$$
= \sup_{F \in \mathcal{F}} \left\{ \mathbb{E}_{P\left(f_v, f_{\mathcal{N}_v^{\mathcal{T}}}\right)} \left[\log \sigma\left(F\left(\mathbf{X}_v, \mathbf{X}_{\mathcal{N}_v^{\mathcal{T}}}\right)\right)\right] \right\}
$$

$$
+ \sup_{F \in \mathcal{F}} \left\{ \mathbb{E}_{P_{f_v}, P_{f_{\mathcal{N}_v^{\mathcal{T}}}}} \left[\log\left(1 - \sigma\left(F\left(\mathbf{X}_v, \mathbf{X}_{\mathcal{N}_{\tilde{v}}^{\mathcal{T}}}\right)\right)\right)\right] \right\},
$$

where, $\sigma(\cdot)$ denotes the sigmoid function. Ultimately, the GAN-like divergence transforms the $f$-divergence into a binary cross-entropy, akin to the objective function used for training the discriminator in GAN Goodfellow et al. (2014). In the aforementioned process of selecting sub-nodes suitable for generating knowledge for the current partition $\mathcal{X}_p$, the above lower bound consists of two components. The first term assesses the effective representational capability of the selected node for its generalized neighborhoods. Considering the close correlation of the definition of generalized neighborhoods with the current partition, it can be regarded as a measure from the embedding perspective of the relevance of the selected node to the knowledge of the current partition. The second term binds the measurement space of relevance with the sampling space based on the affiliation relationship. It gauges the expressive capability of the currently selected node for partition knowledge compared to other nodes in the sampling set. Based on the aforementioned inference, we can generalize it to nodes $u$ belonging to other partitions $\mathcal{X}_q$.

### A.8 The Proof of Theorem 3.3

To determine the form of the function $F(\cdot, \cdot)$, we parametrize $F(\cdot, \cdot)$ using trainable neural networks instead of manual design. The parameterized function is denoted as $F_w(\cdot, \cdot)$, where $w$ generally represents the trainable parameters. In this study, $T_w(\cdot, \cdot)$ has two construction mechanisms based on the partition $\mathcal{X}_i$ to which the selected node belongs and the current partition $\mathcal{X}_p$ where the knowledge generation process is applied. The criteria are as follows:

(1) Intra-partition: Identifying nodes $v$ that efficiently represent the current partition $\mathcal{X}_p$ (i.e., MI between $\mathbf{X}_v$ and $\mathbf{X}_{\mathcal{N}_v^{\mathcal{T}}} = \mathcal{X}_p$) and assigning them higher affinity scores to dominate the weighted knowledge generation process based on the $\Omega_p$.

(2) Inter-partition: Identifying nodes $u$ within other partitions $\mathcal{X}_q$ that potentially represent the current partition effectively (i.e., MI between $\mathbf{X}_u$ and $\mathcal{X}_p$). Meanwhile, node $u$ is required to adhere to well-defined criteria for directed structural information measurement inherited from its corresponding partition to ensure accuracy (i.e., MI between $\mathbf{X}_u$ and $\mathcal{X}_q$). Building upon this foundation, we achieve MI neural estimation between $\mathbf{X}_u$ and $\mathbf{X}_{\mathcal{N}_v^{\mathcal{T}}} = \mathcal{X}_p \cup \mathcal{X}_q$ to obtain efficient affinity scores for $u \in \mathcal{X}_q$. These nodes might not have been correctly assigned to the current partition $\mathcal{X}_p$ initially due to coarse-grained directed structural measurements.

Following these criteria, we reformulate the problem into a fine-grained selection task for nodes contained within two partition roles $\mathcal{X}_p$ and $\mathcal{X}_q$. Building on this, we provide the instantiation of the criterion function $C(\cdot)$, incorporating (1) a model-agnostic digraph learning function $\mathcal{M}$ executed at each tree layer of the HKT, which can leverage some widely used model architectures such as DiGCN Tong et al. (2020a), MagNet Zhang et al. (2021c), HoloNet Koke & Cremers (2023), or be tailored for practical settings; (2) mapping functions $\mathcal{W}_1$ and $\mathcal{W}_2$ dedicated to encoding the currently selected node and its generalized neighborhoods, respectively; (3) two functions $\mathcal{Q}_{intra}$ and $\mathcal{Q}_{inter}$ for generating the final affinity scores based on the encoding results and the current node's partition affiliation. Furthermore, to efficiently encode the generalized neighborhoods, we perform an $l$-step label propagation based on the high-level neighborhood relations $\mathcal{T}_{\mathcal{X}_i}$ in partition $\mathcal{X}_i$ provided by the HKT. The above process based on the current partition $\mathcal{X}_p$ can be formally defined as

$$F_w^{intra} := \mathcal{Q}_{intra}\left(\mathcal{W}_1\left(\mathcal{M}\left(\mathbf{X}_v\right)\right), \mathcal{W}_2\left(\mathcal{M}\left(\mathbf{X}_{\mathcal{N}_v^{\mathcal{T}}}\right)\right)\right),$$

$$F_w^{inter} := \mathcal{Q}_{inter}\left(\mathcal{W}_1\left(\mathcal{M}\left(\mathbf{X}_u\right)\right), \mathcal{W}_2\left(\mathcal{M}\left(\mathbf{X}_{\mathcal{N}_u^{\mathcal{T}}}\right)\right)\right),$$

$$\mathbf{X}_{\mathcal{N}_v^{\mathcal{T}}} = \mathrm{Agg}\left(\hat{\mathbf{X}}_i^l, \forall i \in \mathcal{X}_p\right), \ \mathbf{X}_{\mathcal{N}_u^{\mathcal{T}}} = \mathrm{Agg}\left(\hat{\mathbf{X}}_i^l, \forall i \in \mathcal{X}_p \cup \mathcal{X}_q\right), \tag{20}$$

$$\hat{\mathbf{X}}_i^l = \tau \mathbf{X}_i^0 + (1-\tau) \sum_{j \in \mathcal{T}_{\mathcal{X}_p} \text{ or } j \in \mathcal{T}_{\mathcal{X}_p} \cup \mathcal{T}_{\mathcal{X}_q}} \frac{1}{\sqrt{\tilde{d}_i \tilde{d}_j}} \hat{\mathbf{X}}_i^{l-1}, \ \forall i \in \mathcal{X}_p \text{ or } i \in \mathcal{X}_p \cup \mathcal{X}_q.$$

We adopt the approximate calculation method for the personalized PageRank Klicpera et al. (2019). Meanwhile, we set $\tau = 0.5$ and $l = 5$ by default to capture deep structural information. Due to the small-world phenomenon, we aim to traverse as many nodes as possible within the subgraph through such settings. Moreover, $\mathrm{Agg}(\cdot)$ is a generalized neighborhood representation aggregation function. This function can be implemented through weight-free operations. It is noteworthy that, due to the shared encoding function weights within each partition $\mathcal{X}_i$, the results generated by the neighborhood representation function in partitions with different node quantities must have the same size. In our implementation, considering computational costs, we default to using the weight-free form.

In this manner, the parameterized GAN-like divergence serves as a variational lower bound for the theoretical GAN-like-divergence-based MI between digraph nodes and their generalized neighborhoods. Taking the node $v$ belonging to the current partition $\mathcal{X}_p$ as an example, we obtain the following representation. Similarly, an extension can be applied to nodes $u$ belonging to other partitions $\mathcal{X}_q$.

$$\mathcal{I}_{\mathrm{GAN}}^{(\Omega)}\left(f_v, f_{\mathcal{N}_v^{\mathcal{T}}}\right)$$

$$= \mathcal{D}_{\mathrm{GAN}}\left(P\left(f_v, f_{\mathcal{N}_v^{\mathcal{T}}}\right) \| P_{f_v} \otimes P_{f_{\mathcal{N}_v^{\mathcal{T}}}}\right) \geq \hat{\mathcal{I}}_{\mathrm{GAN}}^{(\Omega)}\left(f_v, f_{\mathcal{N}_v^{\mathcal{T}}}\right)$$

$$= \max_w \left\{ \mathbb{E}_{P\left(f_v, f_{\mathcal{N}_v^{\mathcal{T}}}\right)} \left[\log \sigma\left(F\left(\mathbf{X}_v, \mathbf{X}_{\mathcal{N}_v^{\mathcal{T}}}\right)\right)\right] \right\}$$

$$+ \max_w \left\{ \mathbb{E}_{P_{f_v}, P_{f_{\mathcal{N}_v^{\mathcal{T}}}}} \left[\log\left(1 - \sigma\left(F\left(\mathbf{X}_v, \mathbf{X}_{\mathcal{N}_{\bar{v}}^{\mathcal{T}}}\right)\right)\right)\right] \right\} \tag{21}$$

$$= \max_w \frac{1}{|\Omega|} \sum_{v \in \Omega} \log \sigma\left(F_w\left(\mathbf{X}_v, \mathbf{X}_{\mathcal{N}_v^{\mathcal{T}}}\right)\right)$$

$$+ \max_w \frac{1}{|\Omega|^2} \sum_{(v, \bar{v}) \in \Omega} \log\left(1 - \sigma\left(F_w\left(\mathbf{X}_v, \mathbf{X}_{\mathcal{N}_{\bar{v}}^{\mathcal{T}}}\right)\right)\right).$$

## A.9 ALGORITHM COMPLEXITY ANALYSIS

The complexity of Step 1 is $O\left(h(m \log n + n)\right)$. Notably, as $\mathcal{T}$ tends to be balanced during the structural measurement minimization, height $h$ is approximately $\log n$. Additionally, considering that $m \gg n$, the complexity of Step 1 scales nearly linearly with the number of edges. Subsequently, Step 2 and Step 3 introduce the KD-based training framework. Considering $L$-layer MLP and HKT layer-wise DiGNN, the time complexity can be bound by $O(h(Lmf + Lkn \log nc^2))$. In comparison to Step 1, it is negligible. This is attributed to the random walk and feature transformation can be executed with significantly lower costs due to sparse matrices and parallelism in computation. Moreover, in practice, we can employ a lightweight HKT layer-wise digraph learning to achieve acceleration. Consequently, $O(m)$ in Step 1 remains the primary bottleneck for achieving scalability.

Table 5: The statistical information of the experimental di(graph) benchmark datasets.

| Datasets | #Node | #Features | #Edges | #N Classes | #N Train/Val/Test | #L Train/Val/Test | #Task | Description |
|---|---|---|---|---|---|---|---|---|
| Photo | 7,487 | 745 | 119,043 | 8 | 612/612/5,889 | Undirected | Transductive N | Co-purchase |
| Computers | 13,381 | 767 | 245,778 | 10 | 1100/1100/10651 | Undirected | Transductive N | Co-purchase |
| PPI | 56,944 | 50 | 818,716 | 121 | 4,555/4,555/39,993 | Undirected | Inductive N | Protein |
| Flickr | 89,250 | 500 | 899,756 | 7 | 7,140/7,140/47,449 | Undirected | Inductive N | Image |
| CoraML | 2,995 | 2,879 | 8,416 | 7 | 140/500/2355 | 80%/15%/5% | Node&Link | Citation |
| CiteSeer | 3,312 | 3,703 | 4,591 | 6 | 120/500/2692 | 80%/15%/5% | Node&Link | Citation |
| WikiCS | 11,701 | 300 | 290,519 | 10 | 580/1769/5847 | 80%/15%/5% | Node&Link | Weblink |
| Tolokers | 11,758 | 10 | 519,000 | 2 | 50%/25%/25% | 80%/15%/5% | Node&Link | Crowd-sourcing |
| Empire | 22,662 | 300 | 32,927 | 18 | 50%/25%/25% | 80%/15%/5% | Node&Link | Article Syntax |
| Rating | 24,492 | 300 | 93,050 | 5 | 50%/25%/25% | 80%/15%/5% | Node&Link | Rating |
| Arxiv | 169,343 | 128 | 2,315,598 | 40 | 60%/20%/20% | 80%/15%/5% | Node&Link | Citation |
| Slashdot | 75,144 | 100 | 425,702 | - | - | 80%/15%/5% | Link | Social |
| Epinions | 114,467 | 100 | 717,129 | - | - | 80%/15%/5% | Link | Social |
| WikiTalk | 2,388,953 | 100 | 5,018,445 | - | - | 80%/15%/5% | Link | Co-editor |

## A.10 DATASET DESCRIPTION

We evaluate the performance of our proposed EDEN on 10 digraph and 4 undirected graph benchmark datasets, considering the node-level transductive/inductive semi-supervised classification task and three link-level prediction tasks. The 10 publicly partitioned digraph datasets include 3 citation networks (CoraML, Citeseer, and ogbn-arxiv) in Bojchevski & Günnemann (2018); Hu et al. (2020), 2 social networks (Slashdot and Epinions) in Ordozgoiti et al. (2020); Massa & Avesani (2005), web-link network (WikiCS) in Mernyei & Cangea (2020), crowd-sourcing network (Toloklers) Platonov et al. (2023), syntax network (Empire), rating network (Rating) Platonov et al. (2023), and co-editor network Leskovec et al. (2010). In the transductive scenario, we conduct experiments on two co-purchase networks. In the inductive scenario, we perform experiments on the image relation and the protein interaction networks. The dataset statistics are shown in Table 5 and more descriptions can be found later in this section.

We need to clarify that we are using the directed version of the dataset instead of the one provided by the PyG library (CoraML, CiteSeer)[1], WikiCS paper[2] and the raw data given by the OGB (ogb-arxiv)[3]. Meanwhile, we remove the redundant multiple and self-loop edges to further normalize the 10 digraph datasets. In addition, for Slashdot, Epinions, and WikiTalk, the PyGSD He et al. (2023) library reveals only the topology and lacks the corresponding node features and labels. Therefore, we generate the node features using eigenvectors of the regularised topology. Building upon this foundation, the description of all digraph benchmark datasets is listed below:

**Photo** and **Computers** Shchur et al. (2018) are segments of the Amazon co-purchase graph. Nodes represent goods and edges represent that two goods are frequently bought together. Given product reviews as bag-of-words node features, the task is to map goods to their respective product category.

**PPI** Zeng et al. (2020) stands for Protein-Protein Interaction (PPI) network, where nodes represent protein. If two proteins participate in a life process or perform a certain function together, it is regarded as an interaction between these two proteins. Complex interactions between multiple proteins can be described by PPI networks.

**Flickr** Zeng et al. (2020) dataset originates from the SNAP, they collect Flickr data and generate an undirected graph. Nodes represent images, and edges connect images with common properties like geographic location, gallery, or shared comments. Node features are 500-dimensional bag-of-words representations extracted from the images. The labels are manually merged from the 81 tags into 7 classes.

**CoraML** and **CiteSeer** Bojchevski & Günnemann (2018) are three citation network datasets. In these three networks, papers from different topics are considered nodes, and the edges are citations among the papers. The node attributes are binary word vectors, and class labels are the topics the papers belong to.

---

[1]https://pytorch-geometric.readthedocs.io/en/latest/modules/datasets.html

[2]https://github.com/pmernyei/wiki-cs-dataset

[3]https://ogb.stanford.edu/docs/nodeprop/

**WikiCS** Mernyei & Cangea (2020) is a Wikipedia-based dataset for bench-marking GNNs. The dataset consists of nodes corresponding to computer science articles, with edges based on hyperlinks and 10 classes representing different branches of the field. The node features are derived from the text of the corresponding articles. They were calculated as the average of pre-trained GloVe word embeddings Pennington et al. (2014), resulting in 300-dimensional node features.

**Tolokers** Platonov et al. (2023) is derived from the Toloka crowdsourcing platform Likhobaba et al. (2023). Nodes correspond to tolokers (workers) who have engaged in at least one of the 13 selected projects. An edge connects two tolokers if they have collaborated on the same task. The objective is to predict which tolokers have been banned in one of the projects. Node features are derived from the worker's profile information and task performance statistics.

**Empire** Platonov et al. (2023) is based on the Roman Empire article from the English Wikipedia Lhoest et al. (2021), each node in the graph corresponds to a non-unique word in the text, mirroring the article's length. Nodes are connected by an edge if the words either follow each other in the text or are linked in the sentence's dependency tree. Thus, the graph represents a chain graph with additional connections.

**Rating** Platonov et al. (2023) is derived from the Amazon product co-purchasing network metadata available in the SNAP[4] datasets Leskovec & Krevl (2014). Nodes represent various products, and edges connect items frequently bought together. The task involves predicting the average rating given by reviewers, categorized into five classes. Node features are based on the mean FastText embeddings Grave et al. (2018) of words in the product description. To manage graph size, only the largest connected component of the 5-core is considered.

**ogbn-arxiv** Hu et al. (2020) is a citation graphs indexed by MAG Wang et al. (2020). Each paper comes with a 128-dimensional feature vector obtained by averaging the embeddings of words in its title and abstract. The embeddings of individual words are computed by running the skip-gram model.

**Slashdot** Ordozgoiti et al. (2020) is from a technology-related news website with user communities. The website introduced Slashdot Zoo features that allow users to tag each other as friends or foes. The dataset is a common signed social network with friends and enemies labels. In our experiments, we only consider friendships.

**Epinions** Massa & Avesani (2005) is a who-trust-whom online social network. Members of the site can indicate their trust or distrust of the reviews of others. The network reflects people's opinions of others. In our experiments, we only consider the "trust" relationships.

**WikiTalk** Leskovec et al. (2010) includes all users and discussions from the inception of Wikipedia until January 2008. The network comprises $n = 2,388,953$ nodes, where each node represents a Wikipedia user, and a directed edge from node $v_i$ to node $v_j$ indicates that user $i$ edited user $j$ 's talk page at least once. For our analysis, we extract the largest weakly connected component.

### A.11   COMPARED BASELINES

The baselines we employ are as follows: (1) Directed spatial-based approaches: DGCN Tong et al. (2020b), DIMPA He et al. (2022b), NSTE Kollias et al. (2022), D-HYPR Zhou et al. (2022), and Dir-GNN Rossi et al. (2023); (2) Directed spectral-based approaches: DiGCN Tong et al. (2020a), MagNet Zhang et al. (2021c), MGCZhang et al. (2021a), and HoloNet Koke & Cremers (2023). Furthermore, to verify the generalization of our proposed EDEN, we compare the undirected GNNs in digraphs with coarse undirected transformation (i.e., convert directed edges into undirected edges): GCN Kipf & Welling (2017), GAT Veličković et al. (2018), GCNII Chen et al. (2020), GATv2 Brody et al. (2022), OptBasisGNN Guo & Wei (2023) (OptBG), NAGphormer Chen et al. (2023) (NAG), and AGT Ma et al. (2023). The descriptions of them can be found later in this section. For link-level dataset split, we are aligned with previous work Zhang et al. (2021c); He et al. (2022a; 2023). To alleviate the influence of randomness, we repeat each experiment 10 times to represent unbiased performance and running time (second report). Notably, we present experiment results with various baselines in separate modules, avoiding abundant charts and validating the generalizability of EDEN.

---

[4]https://snap.stanford.edu/

Notably, EDEN can be regarded as a novel digraph learning paradigm or a hot-and-plug online distillation module for prevalent (Di)GNNs. Now, we elaborate on their experimental implementations.

(1) *A new digraph learning paradigm*: Different from the direct application of existing DiGNNs, in the HKT layer-wise distillation process based on HKT, we implement the digraph learning functions of Eq.(7) and Eq.(8) through personalized model design. Specifically, to reduce computational costs, we employ the magnetic Laplacian proposed in MagNet Zhang et al. (2021c) for digraph convolution. Compared to MagNet, EDEN pre-computes $L$ iterations of feature propagation and compresses complex learning processes into simple linear mappings, maximizing training and inference efficiency. Building upon this, a personalized model design for the online distillation process is implemented to achieve end-to-end training.

(2) *A hot-and-plug online distillation module*: Essentially, EDEN serves as a general online distillation framework, introducing a hierarchical knowledge transfer mechanism for existing DiGNNs. In other words, EDEN seamlessly integrates into the HKT layer-wise digraph learning functions (i.e., utilize existing digraph neural architectures as digraph learning function in Eq.(7) and Eq.(8) to generate node embeddings or soft labels) to improve predictions.

**DGCN** Tong et al. (2020b): DGCN proposes the first and second-order proximity of neighbors to design a new message-passing mechanism, which in turn learns aggregators based on incoming and outgoing edges using two sets of independent learnable parameters.

**DIMPA** He et al. (2022b): DIMPA represents source and target nodes separately. However, DIMPA aggregates the neighborhood information within $K$ hops in each layer to further increase the receptive field (RF), and it performs a weighted average of the multi-hop neighborhood information to capture the local network information.

**NSTE** Kollias et al. (2022): NSTE is inspired by the 1-WL graph isomorphism test, which uses two sets of trainable weights to encode source and target nodes separately. Then, the information aggregation weights are tuned based on the parameterized feature propagation process to generate node representations.

**D-HYPR** Zhou et al. (2022): D-HYPR introduces hyperbolic collaborative learning from diverse neighborhoods and incorporates socio-psychological-inspired regularizers. This conceptually simple yet effective framework extends seamlessly to digraphs with cycles and non-transitive relations, showcasing versatility in various downstream tasks.

**Dir-GNN** Rossi et al. (2023): Dir-GNN introduces a versatile framework tailored for heterophilous settings. It addresses edge directionality by conducting separate aggregations of incoming and outgoing edges. Demonstrated to match the expressivity of the directed Weisfeiler-Lehman test, Dir-GNN outperforms conventional MPNNs in accurately modeling digraphs.

**DiGCN** Tong et al. (2020a): DiGCN notices the inherent connections between graph Laplacian and stationary distributions of PageRank, it theoretically extends personalized PageRank to construct real symmetric Digraph Laplacian. Meanwhile, DiGCN uses first-order and second-order neighbor proximity to further increase RF.

**MagNet** Zhang et al. (2021c): MagNet utilizes complex numbers to model directed information, it proposes a spectral GNN for digraphs based on a complex Hermitian matrix known as the magnetic Laplacian. Meanwhile, MagNet uses additional trainable parameters to combine the real and imaginary filter signals separately to achieve better prediction performance.

**MGC** Zhang et al. (2021a): MGC introduces the magnetic Laplacian, a discrete operator with the magnetic field, which preserves edge directionality by encoding it into a complex phase with an electric charge parameter. By adopting a truncated variant of PageRank, it designs and builds a low-pass filter for homogeneous graphs and a high-pass filter for heterogeneous graphs.

**HoloNet** Koke & Cremers (2023): HoloNet demonstrates that spectral convolution can extend to digraphs. By leveraging advanced tools from complex analysis and spectral theory, HoloNet introduces spectral convolutions tailored for digraphs.

**GCN** Kipf & Welling (2017): GCN is guided by a localized first-order approximation of spectral graph convolutions. This model's scalability is directly proportional to the number of edges, and it learns intermediate representations in hidden layers that capture both the structure and node features.

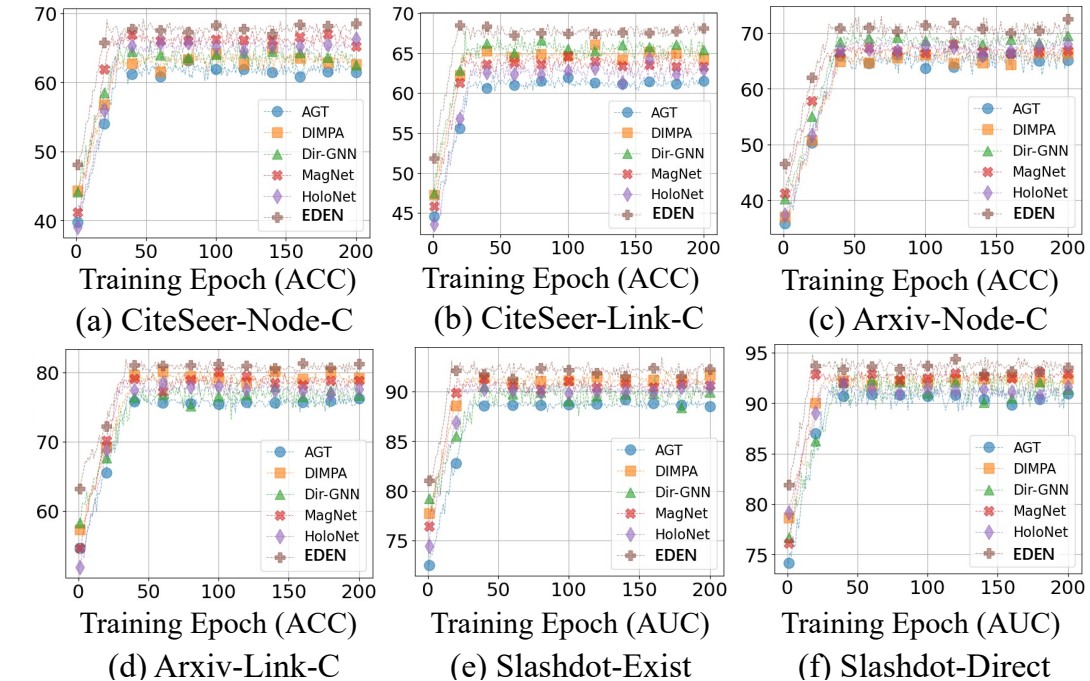

Figure 6: Convergence curves on node- and link-level tasks.

**GCNII** Chen et al. (2020) incorporates initial residual and identity mapping. Theoretical and empirical evidence is presented to demonstrate these techniques alleviate the over-smoothing issue.

**GAT** Veličković et al. (2018) utilizes attention mechanisms to quantify the importance of neighbors for message aggregation. This strategy enables implicitly specifying different weights to different nodes in a neighborhood, without depending on the graph structure upfront.

**GATv2** Brody et al. (2022) introduces a variant with dynamic graph attention mechanisms to improve GAT. This strategy provides better node representation capabilities and enhanced robustness when dealing with graph structure as well as node attribute noise.

**OptBasisGNN** Guo & Wei (2023): OptBasisGNN revolutionizes GNNs by redefining polynomial filters. It dynamically learns suitable polynomial bases from training data, addressing fundamental adaptability. OptBasisGNN innovatively addresses the challenge of determining the optimal polynomial basis for a specific graph and signal, showcasing its effectiveness in extensive experiments.

**NAGphormer** Chen et al. (2023) treats each node as a sequence containing a series of tokens. For each node, NAGphormer aggregates the neighborhood features from different hops into different representations.

**AGT** Ma et al. (2023) consists of a learnable centrality encoding strategy and a kenneled local structure encoding mechanism to extract structural patterns from the centrality and subgraph views to improve node representations for the node-level downstream tasks.

## A.12 HYPERPARAMETER SETTINGS

The hyperparameters in the baseline models are set according to the original paper if available. Otherwise, we perform a hyperparameter search via the Optuna Akiba et al. (2019). For our proposed EDEN, during the topology-aware coarse-grained HKT construction, we perform a grid search in the interval $[3, 10]$ to determine the height of HKT. In the feature-oriented fine-grained HKT correction, a grid search is conducted in the interval $[1, 2]$ to obtain the optimal $\kappa$, deciding the knowledge reception field when generating parent node representations for the current partition. For random walk-based leaf prediction, we search in the interval $[0, 1]$ based on node-level or link-level downstream tasks to determine the optimal walking probability. Additionally, within the same interval, we search to determine the hyperparameter $\alpha$ for knowledge distillation loss, ensuring optimal convergence.

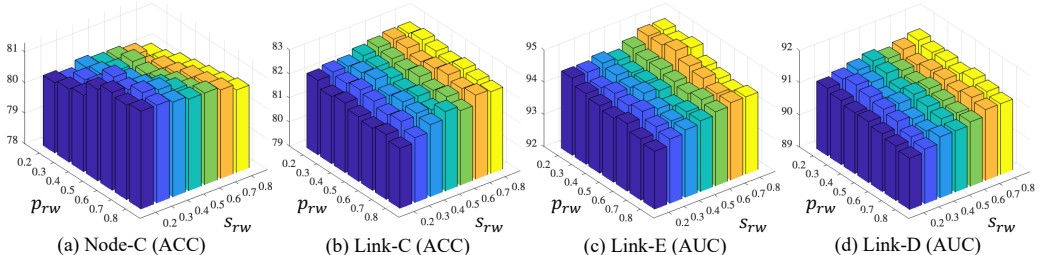

Figure 7: The sensitive analysis of HKT-based random walk under Tolokers.

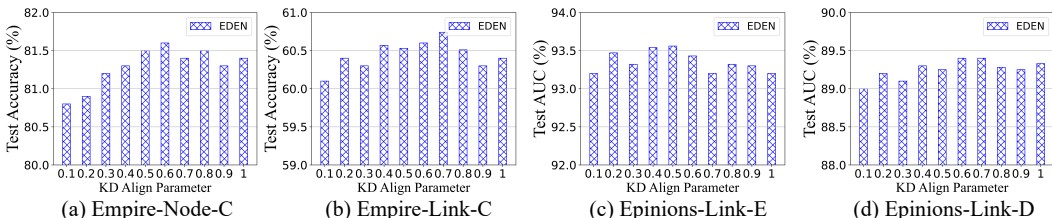

Figure 8: The sensitive analysis of KD loss factor.

## A.13 EXPERIMENT ENVIRONMENT

The experiments are conducted on Intel(R) Xeon(R) Gold 6230R CPU @ 2.10GHz, NVIDIA GeForce RTX 3090 with 24GB memory, and CUDA 11.8. The operating system is Ubuntu 18.04.6 with 768GB of memory. As for software versions we use Python 3.9 and Pytorch 1.11.0.

## A.14 EXTEND EXPERIMENTAL RESULTS

**Convergence Analysis**. To supplement answer **Q3**, we first present the convergence curves in Fig. 6, where we observe that EDEN exhibits higher initial performance and more stable convergence. For instance, in the Node-C for the CiteSeer, EDEN nearly reaches converged performance by the 25th epoch and maintains stability throughout the subsequent training process. Notably, various link-level downstream tasks, benefiting from a larger number of training samples, exhibit smoother optimization curves and more stable predictive performances compared to node-level classification tasks.

**Hyperparameter Analysis**. To provide a comprehensive analysis of the robustness of EDEN from the perspective of hyperparameter sensitivity, we supplement the experimental results in Fig.7 with the outcomes of HKT-based random walk sampling for leaf-centric prediction, considering various probabilities of transitioning between different identity nodes (i.e., parents, siblings, and children). Notably, we do not discuss the sampling probability regarding children separately. This is because their main role is to provide return probabilities in the random walk process to yield richer sampling sequences, without explicitly indicating the identity of the next node to visit. Before giving our analysis, we first revisit the key insights introduced in Sec.3.3: (1) For node-level downstream tasks, it's preferable to sample the parent of the current leaf node to offer a rich high-level representation of the current label class. (2) For link-level downstream tasks, it's preferable to sample the siblings of the current leaf node to provide topologically relevant contextual insights at the same level. Based on the experimental results, we observe that for Node-C, larger values of $p_{rw}$ and smaller values of $s_{rw}$ yield better predictive performance, whereas for the three distinct link-level downstream tasks, smaller values of $p_{rw}$ and larger values of $s_{rw}$ are preferable. This validates our aforementioned assertions and provides an empirical reference for selecting hyperparameters when practically applying EDEN.

Additionally, in Fig. 8, we provide insights into how varying the coefficient $\alpha$ in the $\alpha$-flexible KD loss impacts the optimization process, reflected in the final predictive performance. According to our experimental results, in most cases, EDEN should prioritize the KD process during end-to-end optimization. This is because the node-adaptive trainable knowledge generation and transfer processes ensure high-quality KD, thereby positively influencing downstream task predictions. Notably, smaller values of $\alpha$ perform better in edge existence problems. This is because the cross-entropy loss function, used to provide supervision, aids significantly in coarser-grained existence problems, while finer-grained issues like directionality and classification often benefit more from data-driven high-quality knowledge. In a nutshell, we recommend smaller $\alpha$ values for edge existence problems and larger $\alpha$ values for other tasks, followed by manual adjustments based on practical performance.

Table 7: Model performance (%) in three directed link-level downstream tasks.

| Datasets (→) | Slashdot | | | | | Epinions | | | | |
|---|---|---|---|---|---|---|---|---|---|---|
| Tasks (→) | Exist | | Direct | | Link-C | Exist | | Direct | | Link-C |
| Models (↓) | AUC | AP | AUC | AP | ACC | AUC | AP | AUC | AP | ACC |
| GCNII | 88.6±0.1 | 88.4±0.0 | 90.3±0.1 | 90.4±0.1 | 84.0±0.1 | 91.3±0.1 | 91.3±0.0 | 85.9±0.2 | 86.3±0.1 | 82.7±0.1 |
| GATv2 | 88.2±0.2 | 88.5±0.1 | 90.6±0.1 | 90.4±0.1 | 83.7±0.3 | 91.8±0.2 | 91.6±0.1 | **85.5±0.1** | 85.9±0.1 | 83.0±0.2 |
| AGT | 88.7±0.2 | 88.6±0.1 | 90.1±0.0 | 90.5±0.1 | 83.8±0.2 | 91.5±0.2 | 91.4±0.2 | 85.7±0.2 | 86.2±0.2 | 83.4±0.1 |
| DGCN | 90.3±0.1 | 90.1±0.0 | 92.2±0.1 | 92.4±0.1 | 85.5±0.2 | 92.2±0.1 | 92.5±0.0 | 87.8±0.1 | 87.5±0.2 | 83.6±0.2 |
| DIMPA | 90.5±0.1 | 90.7±0.1 | 92.4±0.2 | 92.1±0.1 | 85.6±0.1 | 92.5±0.1 | 92.6±0.1 | 87.9±0.1 | 88.2±0.1 | 83.5±0.1 |
| D-HYPR | 90.3±0.0 | 90.6±0.1 | 92.2±0.1 | 91.9±0.0 | 85.4±0.1 | 92.8±0.1 | 92.4±0.1 | 88.2±0.1 | 88.3±0.0 | 83.7±0.2 |
| DiGCN | 90.4±0.1 | 90.5±0.1 | 92.1±0.1 | 92.0±0.1 | 85.2±0.1 | 92.4±0.1 | 92.7±0.1 | 88.0±0.1 | 87.8±0.1 | 83.6±0.1 |
| HoloNet | 90.2±0.1 | 90.3±0.0 | 91.8±0.1 | 92.0±0.0 | 85.1±0.1 | 92.6±0.1 | 92.5±0.0 | 88.1±0.1 | 88.2±0.0 | 84.0±0.1 |
| EDEN | **91.8±0.1** | **92.0±0.0** | **93.3±0.1** | **93.1±0.0** | **87.1±0.2** | **93.5±0.1** | **93.7±0.0** | **89.4±0.1** | **89.8±0.0** | **85.7±0.1** |

**Comprehensive Results**.

To present comprehensive experimental findings, this section includes additional results (Table 6, Table 7, Table 8, and Table 9) that couldn't be fully showcased in the main text due to space limitations. These additional experimental results, consistent with the trends presented in the main text, further substantiate our claims in Sec. 4. Notably, to provide a more thorough assessment, we introduce two additional evaluation metrics, Area Under Curve (AUC) and Average Precision (AP), alongside the commonly known Accuracy (ACC). We default to using AUC

Table 6: Test accuracy (%) in directed Node-C.

| Models | CoraML | CiteSeer | WikiCS | Tolokers | Empire | Rating | Arxiv |
|---|---|---|---|---|---|---|---|
| GCN | 80.6±0.4 | 62.1±0.4 | 78.3±0.2 | 78.0±0.1 | 75.8±0.5 | 42.5±0.4 | 65.2±0.2 |
| GAT | 80.7±0.6 | 62.6±0.6 | 78.2±0.3 | 78.4±0.2 | 77.8±0.8 | 42.9±0.5 | 65.9±0.3 |
| OptBG | 81.0±0.5 | 63.2±0.4 | 78.5±0.2 | 78.6±0.2 | 78.0±0.5 | 43.2±0.4 | 66.3±0.3 |
| NAG | 81.4±0.7 | 62.7±0.5 | 78.6±0.3 | 78.4±0.4 | 77.5±0.9 | 43.1±0.6 | 66.5±0.4 |
| NSTE | 82.2±0.5 | 64.3±0.7 | 79.0±0.3 | 79.3±0.3 | 78.9±0.6 | 44.7±0.6 | 67.2±0.4 |
| Dir-GNN | 82.6±0.6 | 64.0±0.6 | 79.1±0.4 | 79.1±0.3 | 79.1±0.5 | 45.0±0.5 | 67.4±0.3 |
| MGC | 82.3±0.4 | 63.9±0.5 | 78.8±0.2 | 79.0±0.2 | 78.6±0.4 | 44.8±0.4 | 67.0±0.2 |
| EDEN | **84.6±0.5** | **65.8±0.6** | **81.4±0.3** | **81.3±0.2** | **81.1±0.6** | **46.3±0.4** | **69.7±0.3** |

and AP in the evaluation of the link prediction tasks, and ACC to evaluate the predictive performance of node-level tasks. Regarding the experimental results of Dir-GNN and HoloNet on the Empire dataset, we would like to clarify that we ensured a fair comparison by using a class-balanced dataset split instead of the pre-split datasets used in Dir-GNN and HoloNet.

**AUC** stands as a comprehensive metric for evaluating binary classification models. Quantifying the area beneath the ROC curve, it provides a global assessment of the model's ability to discriminate between positive and negative instances. AUC is particularly valuable in scenarios with imbalanced datasets, as it remains insensitive to variations in class distribution. Its utility extends to model comparison, offering insights into performance variations across different decision thresholds.

**AP** involves ranking predictions by their confidence scores, typically probabilities, from highest to lowest, and calculating precision and recall at each threshold. These metrics are used to construct a precision-recall curve, which plots precision values as a function of recall. AP itself is computed as the weighted mean of precision achieved at each threshold, where the weights are the increments in recall from the previous thresholds. This approach allows AP to summarize the area under the precision-recall curve, providing a single-figure measure of model performance that encapsulates both the accuracy and the ranking of the positive predictions. Higher AP values indicate a model that not only predicts the positive class accurately but also ranks those predictions highly, thus demonstrating high precision and recall across the board.

Table 8: Link-C ACC and others AUC (%) in three directed link-level downstream tasks.

| Datasets | Tasks | GCN | GAT | OptBG | NAG | NSTE | Dir-GNN | MGC | HoloNet | EDEN |
|---|---|---|---|---|---|---|---|---|---|---|
| CoraML | Existence | 83.26±0.18 | 83.96±0.25 | 83.55±0.16 | 84.32±0.20 | 87.94±0.18 | 88.15±0.21 | 87.86±0.20 | 87.80±0.24 | **90.84±0.19** |
| | Direction | 82.73±0.32 | 84.25±0.54 | 83.46±0.40 | 85.39±0.47 | 90.74±0.54 | 91.08±0.45 | 89.10±0.62 | 89.83±0.57 | **92.36±0.48** |
| | Link-C | 69.80±0.45 | 70.67±0.52 | 70.54±0.60 | 71.04±0.56 | 72.79±0.42 | 73.11±0.49 | 72.82±0.60 | 72.74±0.56 | **75.18±0.54** |
| CiteSeer | Existence | 75.60±0.34 | 76.27±0.28 | 75.85±0.29 | 76.94±0.40 | 79.80±0.42 | 79.65±0.34 | 79.46±0.29 | 79.32±0.30 | **82.24±0.37** |
| | Direction | 72.32±0.75 | 73.46±0.53 | 72.96±0.68 | 73.88±0.47 | 88.35±0.68 | 88.64±0.57 | 88.47±0.41 | 88.76±0.48 | **90.56±0.40** |
| | Link-C | 61.74±0.83 | 62.46±0.72 | 62.29±0.75 | 62.86±0.65 | 64.16±0.48 | 64.35±0.43 | 63.88±0.50 | 63.94±0.36 | **66.73±0.57** |
| WikiCS | Existence | 90.67±0.07 | 91.15±0.14 | 90.43±0.10 | 91.08±0.14 | 91.60±0.08 | 91.38±0.11 | 91.13±0.05 | 91.28±0.09 | **92.84±0.12** |
| | Direction | 85.26±0.37 | 85.61±0.29 | 85.40±0.32 | 85.75±0.35 | 87.28±0.25 | 87.12±0.30 | 87.33±0.17 | 87.24±0.26 | **90.08±0.24** |
| | Link-C | 78.71±0.15 | 79.08±0.19 | 78.84±0.23 | 79.42±0.22 | 81.83±0.19 | 81.67±0.14 | 81.47±0.20 | 81.26±0.18 | **83.45±0.21** |
| Tolokers | Existence | 91.90±0.09 | 92.23±0.14 | 92.08±0.09 | 92.19±0.11 | 93.03±0.14 | 93.48±0.11 | 93.69±0.10 | 93.84±0.08 | **94.93±0.10** |
| | Direction | 87.68±0.13 | 87.57±0.08 | 88.28±0.11 | 88.97±0.09 | 89.42±0.10 | 89.65±0.08 | 89.92±0.07 | 89.76±0.11 | **91.52±0.12** |
| | Link-C | 77.54±0.09 | 77.85±0.14 | 78.20±0.12 | 78.49±0.13 | 80.28±0.07 | 80.46±0.10 | 80.83±0.08 | 80.51±0.12 | **82.67±0.13** |
| Empire | Existence | 62.51±0.67 | 62.93±0.81 | 63.14±0.75 | 63.85±0.80 | 66.35±0.35 | 66.28±0.42 | 65.99±0.32 | 65.86±0.46 | **68.81±0.41** |
| | Direction | 48.60±0.95 | 49.77±0.87 | 49.82±0.93 | 50.16±0.84 | 53.87±0.42 | 53.94±0.40 | 53.58±0.37 | 53.79±0.45 | **55.60±0.48** |
| | Link-C | 52.56±0.86 | 53.02±0.99 | 52.84±1.01 | 53.12±1.17 | 58.69±0.44 | 58.62±0.45 | 58.09±0.31 | 58.33±0.35 | **60.74±0.39** |
| Rating | Existence | 73.48±0.45 | 73.95±0.57 | 73.60±0.52 | 75.26±0.43 | 76.91±0.20 | 77.48±0.29 | 77.21±0.18 | 77.12±0.26 | **79.52±0.27** |
| | Direction | 78.54±0.32 | 78.81±0.41 | 78.90±0.36 | 79.42±0.35 | 82.85±0.27 | 83.46±0.30 | 83.68±0.21 | 83.30±0.33 | **85.19±0.29** |
| | Link-C | 58.63±0.46 | 58.79±0.50 | 58.60±0.64 | 59.13±0.37 | 63.64±0.28 | 64.23±0.39 | 64.28±0.25 | 64.32±0.32 | **66.37±0.35** |
| Arxiv | Existence | 82.04±0.15 | 81.87±0.19 | 82.24±0.17 | 82.44±0.16 | 84.82±0.23 | 85.37±0.19 | 84.70±0.28 | 85.25±0.20 | **87.24±0.23** |
| | Direction | 88.56±0.16 | 88.71±0.20 | 88.94±0.21 | 89.10±0.22 | 93.34±0.14 | 93.62±0.17 | 93.27±0.11 | 93.40±0.15 | **94.48±0.16** |
| | Link-C | 74.70±0.17 | 74.53±0.16 | 74.93±0.20 | 75.05±0.18 | 78.63±0.17 | 78.89±0.15 | 78.70±0.18 | 78.93±0.21 | **80.16±0.21** |

Table 9: Link-C ACC and others AUC (%) in three directed link-level downstream tasks.

| Datasets | Tasks | GCNII | GATv2 | AGT | DGCN | DIMPA | D-HYPR | DiGCN | MagNet | EDEN |
|---|---|---|---|---|---|---|---|---|---|---|
| CoraML | Existence | 84.01±0.22 | 84.58±0.33 | 84.50±0.24 | 87.65±0.20 | 88.06±0.20 | 87.99±0.24 | 87.65±0.28 | 88.05±0.21 | **90.84±0.19** |
| | Direction | 83.25±0.36 | 84.94±0.60 | 85.57±0.51 | 90.43±0.49 | 90.88±0.50 | 90.94±0.54 | 89.75±0.71 | 90.83±0.49 | **92.36±0.48** |
| | Link-C | 70.43±0.55 | 71.24±0.58 | 71.23±0.47 | 72.55±0.48 | 72.86±0.55 | 72.91±0.38 | 72.53±0.56 | 72.96±0.42 | **75.18±0.54** |
| CiteSeer | Existence | 76.24±0.46 | 76.86±0.35 | 76.72±0.38 | 79.65±0.49 | 79.65±0.38 | 79.84±0.29 | 79.32±0.33 | 79.80±0.24 | **82.24±0.37** |
| | Direction | 72.95±0.82 | 74.08±0.56 | 73.76±0.65 | 88.12±0.73 | 88.42±0.70 | 88.75±0.63 | 88.19±0.38 | 88.67±0.45 | **90.56±0.40** |
| | Link-C | 62.37±0.88 | 63.21±0.78 | 62.53±0.57 | 64.02±0.56 | 64.21±0.43 | 64.30±0.37 | 63.92±0.59 | 64.03±0.40 | **66.73±0.57** |
| WikiCS | Existence | 90.98±0.10 | 91.48±0.20 | 90.87±0.18 | 91.24±0.10 | 91.53±0.10 | 91.58±0.11 | 91.49±0.13 | 91.52±0.12 | **92.84±0.13** |
| | Direction | 85.84±0.49 | 86.95±0.32 | 85.65±0.42 | 86.88±0.33 | 87.26±0.29 | 87.35±0.34 | 87.38±0.21 | 87.40±0.18 | **90.08±0.24** |
| | Link-C | 79.28±0.25 | 79.64±0.25 | 79.26±0.29 | 81.12±0.16 | 81.33±0.12 | 81.50±0.19 | 81.66±0.24 | 81.63±0.11 | **83.45±0.21** |
| Tolokers | Existence | 92.31±0.10 | 92.46±0.18 | 92.22±0.08 | 92.41±0.15 | 93.78±0.15 | 93.75±0.14 | 93.42±0.12 | 93.62±0.10 | **94.93±0.10** |
| | Direction | 88.14±0.16 | 88.27±0.10 | 89.12±0.07 | 88.92±0.12 | 89.90±0.11 | 89.94±0.10 | 89.68±0.09 | 89.83±0.09 | **91.52±0.12** |
| | Link-C | 78.10±0.11 | 78.29±0.19 | 78.72±0.15 | 79.74±0.08 | 80.84±0.09 | 80.79±0.09 | 80.52±0.10 | 80.78±0.8 | **82.67±0.13** |
| Empire | Existence | 63.37±0.72 | 63.78±0.90 | 63.92±0.74 | 65.67±0.40 | 66.28±0.32 | 66.31±0.35 | 66.39±0.43 | 66.27±0.34 | **68.81±0.41** |
| | Direction | 49.56±0.90 | 50.64±0.76 | 50.38±0.70 | 53.26±0.32 | 53.92±0.36 | 53.87±0.42 | 53.91±0.50 | 53.84±0.39 | **55.60±0.48** |
| | Link-C | 53.41±0.90 | 54.13±0.84 | 53.43±0.99 | 58.05±0.38 | 58.56±0.49 | 58.64±0.48 | 58.64±0.54 | 58.56±0.26 | **60.74±0.39** |
| Rating | Existence | 74.68±0.54 | 74.83±0.64 | 75.08±0.33 | 76.64±0.24 | 76.84±0.22 | 77.39±0.32 | 77.30±0.29 | 77.31±0.19 | **79.52±0.27** |
| | Direction | 79.32±0.41 | 79.65±0.42 | 79.56±0.37 | 82.34±0.33 | 82.91±0.24 | 83.58±0.29 | 83.62±0.33 | 83.56±0.27 | **85.19±0.29** |
| | Link-C | 59.95±0.62 | 60.27±0.58 | 59.37±0.40 | 63.28±0.23 | 63.78±0.30 | 64.33±0.36 | 64.28±0.40 | 64.32±0.30 | **66.37±0.35** |
| Arxiv | Existence | 83.14±0.23 | 82.54±0.31 | 82.21±0.18 | 84.40±0.19 | 85.19±0.21 | 85.13±0.23 | 85.02±0.31 | 85.29±0.19 | **87.24±0.23** |
| | Direction | 89.20±0.27 | 89.13±0.29 | 89.47±0.28 | 93.05±0.16 | 93.41±0.19 | 93.24±0.20 | 93.18±0.25 | 93.37±0.14 | **94.48±0.16** |
| | Link-C | 75.97±0.21 | 75.60±0.18 | 75.29±0.15 | 78.24±0.25 | 78.90±0.20 | 78.74±0.14 | 78.69±0.26 | 78.97±0.23 | **80.16±0.21** |

