# OpenReview forum: "Entropy-driven Data Knowledge Distillation in Digraph Representation Learning"
_ICLR.cc/2025/Conference — ICLR 2025 Conference Withdrawn Submission_

### Official Review · Reviewer_8ssE · 2024-11-03

**Soundness:** 2
**Presentation:** 2
**Contribution:** 2
**Rating:** 5
**Confidence:** 3

**Summary:**

The paper introduces Entropy-driven Digraph knowlEdge distillatioN (EDEN), a data-centric framework for digraph neural network (DiGNN) models. Recognizing the limitations of current DiGNNs in fully leveraging the rich data encoded in directed graphs, EDEN aims to enhance the encoding capabilities of these models by focusing on the relationships between directed topologies and node features. It constructs a hierarchical knowledge tree based on directed structural measurements, which guides the knowledge flow through mutual information quantification, enabling a tree layer-wise knowledge distillation process. It demonstrates strong performance across diverse datasets, but its complexity presents scalability challenges for very large graphs.

**Strengths:**

1.This paper is the first attempt to introduce the data-centric online knowledge distillation  specifically designed to enhance digraph representation learning.

2.The authors propose a unified framework that emphasizes data-centric learning, employing a hierarchical tree structure that integrates both topological and feature-based information.

3.The framework demonstrates strong performance across a range of datasets.

**Weaknesses:**

1.The paper offers an intriguing perspective; however, the writing could benefit from further refinement to improve clarity and logical flow. For instance, the motivation for the study and the methods proposed to address existing limitations are not clearly articulated. While the authors claim to "explore the potential correlations between complex directed topology and node profiles," the paper does not explicitly demonstrate this correlation or detail why the information is helpful.

2.The transitions between sections and subsections could be smoother, as some concepts are introduced without sufficient context or clear definitions. For example, terms like "teacher node" and "student node" are mentioned without adequate background—leaving it unclear why knowledge transfer is necessary in this digraph setting and how it contributes to the model's goals. Additional background information would help to clarify these concepts and their relevance within the framework.

3.Certain notations lack clear definitions, making them difficult to follow. For instance, symbols like ${d}\_{v}^{in/out}$ seem to imply ​  ${d}\_{v}^{in}$ or ${d}\_{v}^{out}​$, but without proper introduction, their meanings remain ambiguous. Similarly, the notation Q appears in various equations yet is not clearly defined, which can lead to confusion. Improved clarity in notation would enhance the paper's readability and comprehension.

**Questions:**

please refer to the weekness.

---

### Official Review · Reviewer_KzDR · 2024-11-03

**Soundness:** 3
**Presentation:** 2
**Contribution:** 2
**Rating:** 3
**Confidence:** 4

**Summary:**

To address the issue that existing DiGNNs fail to comprehensively delve into the abundant data knowledge concealed in the digraphs. This paper proposes entropy-driven digraph knowledge distillation (EDEN), which is a model-agnostic hot-andplug data online knowledge distillation module to fully leverage informative digraphs. The experiments demonstrates the superior performance of EDEN on 14 digraph datasets and across 4 downstream tasks. The results demonstrate that EDEN attains SOTA performance and exhibits strong improvement.

**Strengths:**

1. EDEN achieves data online KD for empowering digraph representation learning.

2. The EDEN method proposed in the paper is very described.

**Weaknesses:**

1. This paper is not well structured and it is suggested to separate subsection 2.2, 2.3 and 2.4 from the PRELIMINARIES section.

2. The issue the article is trying to solve is not clearly described in the abstract.

3. Experimentation is insufficient and it is recommended that the baseline methodology be added for 2024 year.

4. There are some problems with the article description and please double check the full paper.

**Questions:**

See the Weaknesses.

---

### Official Review · Reviewer_CDXP · 2024-11-03

**Soundness:** 3
**Presentation:** 2
**Contribution:** 4
**Rating:** 5
**Confidence:** 2

**Summary:**

This paper introduces a novel approach to digraph representation learning via a framework named EDEN (Entropy-driven Digraph knowlEdge distillatioN). The authors aim to leverage entropy-based methods to extract data knowledge from directed graphs (digraphs), addressing the limitations of current directed Graph Neural Networks (DiGNNs). EDEN is presented as both a new data-centric paradigm and a model-agnostic module that distills hierarchical knowledge from the digraph’s structure and node features. The process uses a knowledge tree that represents the hierarchical topology and enables layer-wise knowledge distillation through mutual information metrics. Experimental results demonstrate that EDEN consistently achieves state-of-the-art performance.

**Strengths:**

1.Novel Perspective: The paper proposes an entropy-driven framework that adopts a data-centric approach to digraph knowledge distillation, offering new insights into digraph representation.

2.Flexible and Model-agnostic: EDEN is adaptable to existing DiGNNs as a plug-and-play module, enhancing predictions without modifying model architectures.

3.Comprehensive Evaluation: The authors conduct extensive experiments, evaluating the framework on various tasks and datasets to validate its effectiveness.

**Weaknesses:**

1.Despite the motivation being clear, additional intuitive explanations would help to better clarify the necessity of the proposed method.

2.Some terms and concepts, such as the hierarchical knowledge tree and true structure, are introduced without sufficient initial clarification, which could hinder the readability.

**Questions:**

1.How does EDEN manage the potential trade-off between depth in hierarchical representation and computational efficiency on large-scale graphs?

2.Are there any specific limitations observed in the experiments when using EDEN in undirected scenarios, particularly where directional information is absent?

---

### Official Review · Reviewer_3G9e · 2024-11-04

**Soundness:** 3
**Presentation:** 2
**Contribution:** 3
**Rating:** 5
**Confidence:** 3

**Summary:**

This paper introduces a data-centric digraph online knowledge distillation framework named EDEN. The framework utilizes directed structural measurement to construct a hierarchical knowledge tree (HKT). Furthermore, HKT is refined based on the mutual information of node features, which enhances hierarchical and personalized knowledge transfer. Extensive experiments indicate the effectiveness of the proposed method.

**Strengths:**

1. The paper is novel and includes some theoretical analysis.
2. EDEN consistently outperforms other directed GNNs in both node-level classification and link-level prediction tasks.
3. The framework shows promising results on large-scale networks.

**Weaknesses:**

1. The implementation steps of the proposed method appear overly complex, which may hinder the generalizability of the framework.
2. The presentation could be improved to make the work more accessible to researchers who may not have a strong background in this area. As it stands, reproducing the method based on the paper alone could be challenging.

**Questions:**

1. Could the authors elaborate on the significance of disentangling complex directed structural patterns and node profiles? Are there existing studies that have made noteworthy progress in this area?
2. The authors mention that digraphs possess more complex feature knowledge, but the experiments primarily involve homophily networks. How does the method perform on heterophily networks? This would be crucial for assessing the generalizability of a data-centric digraph learning framework.

---

### Official Review · Reviewer_mtPZ · 2024-11-04

**Soundness:** 3
**Presentation:** 1
**Contribution:** 2
**Rating:** 5
**Confidence:** 4

**Summary:**

This paper focuses on the topic of representation learning on directed graphs. The authors consider approaching this problem by properly modeling the underlying generation process of these graphs. They do so through the use of structural measures to estimate the hierarchical knowledge tree (i.e., the "true foundation" of the graph). The core of this approach is a data distillation method that transfers knowledge from a parent node to its children. The prediction for the downstream task is done through the use of a random-walk based technique. They show promising results for node classification and link prediction on multiple datasets.

**Strengths:**

1. The design and motivation of the framework is interesting and unique from earlier works. This is welcome.
2. The authors report a steady performance gain when using EDEN.
3. They demonstrate the efficiency of EDEN, as it doesn't have a large negative impact on runtime.

**Weaknesses:**

1. I found the writing to be quite dense. Many technical definitions without proper context or explanation. For example, it's hard to understand why the structural measurements (Eqs 1-3) are formulated in the way that they are. E.g., Why can Eq 2 better capture higher-order structures? Why can Eq. 3 better consider complex hierarchical structures? This criticism applies to many of the equations in the paper (see the questions for more). While the notation itself is introduced, no higher-level or intuitive explanation is given. This makes it hard to follow and understand. While I understand that the appendix does give a more detailed explanation of many concepts, the reader shouldn't be obliged to refer to it to understand the paper itself.

2. The ablation study (Table 4) seems to suggests limited impact of the "Diverse Knowledge" and "Personalized Transfer" modules. The performance w/o them seems to be roughly within variance for each dataset. In general, the one consistent component that helps performance is the KD loss. These results hurt the contribution presented by some of the components.

3. I'm concerned about the impact of the random walk based prediction on the efficiency. Enumerating random walks for graph learning is a well-known bottleneck. This is especially true on larger graphs. I'd appreciate a more thorough comparison on the running time breakdown by datasets of different sizes. Furthermore what's the maximum length random walk you consider? My concern is that for large/dense graphs it'll be very inefficient to go beyond short random walks.

**Questions:**

1. Can you explain the intuitiions behind Eqs 1, 2, 3 and why they can capture certain types of information?
2. Can you give a high-level understanding of how Eq. 7 is derived from the previous theorems? Why is it reduced to this equation? What goal can it achieve and how does it align with the theorems?
3. I don't entirely understand the need for Eq 8? Why should be interested in the inter-partition information? The theorems only discuss the intra-partition case. I'm not sure I understand the explanation on lines 317-318.
4. Why is knowledge only transferred from parents to children? It seems strange to me as I find it hard to believe that the children of some nodes
5. What datasets are the efficiency analysis (Fig 3) run on? Is it averaged across datasets or just one? I think it would be interesting to see the comparison across datasets of different sizes, to see if the same pattern holds true for smaller and larger graphs.
6. What value do you use for the maximum random walk length?

---

### Note · Authors · 2024-11-13

I have read and agree with the venue's withdrawal policy on behalf of myself and my co-authors.